



# Ice-nucleating particle concentration impacts cloud properties over Dronning Maud Land, East Antarctica, in COSMO-CLM²

Florian Sauerland[1], Niels Souverijns[2], Anna Possner[3], Heike Wex[4], Preben Van Overmeiren[5], Alexander Mangold[6], Kwinten Van Weverberg[7, 8], and Nicole van Lipzig[1]

[1]Department of Earth and Environmental Sciences, KU Leuven, Leuven, Belgium
[2]Environmental Intelligence Unit, Flemish Institute for Technological Research (VITO), Mol, Belgium
[3]Institute for Atmosphere and Environment, Goethe-Universität Frankfurt, Frankfurt am Main, Germany
[4]Department of Atmospheric Microphysics, Leibniz-Institut für Troposphärenforschung, Leipzig, Germany
[5]Department of Green Chemistry and Technology, Ghent University, Ghent, Belgium
[6]Scientific Service Observations, Royal Meteorological Institute of Belgium, Brussels, Belgium
[7]Department of Geography, Ghent University, Ghent, Belgium
[8]Meteorological and Climatological Research Unit, Royal Meteorological Institute of Belgium, Brussels, Belgium

**Correspondence:** Florian Sauerland (florian.sauerland@kuleuven.be)

**Abstract.**

Ice-nucleating particles (INPs) have an important function in the freezing of clouds, but are rare in East Antarctica with concentrations between $6 \times 10^{-6}\,\mathrm{L}^{-1}$ and $5 \times 10^{-3}\,\mathrm{L}^{-1}$ observed at the Belgian Princess Elisabeth Station. These low concentrations offer a possible explanation for the occurrence of supercooled liquid water in clouds observed using the station's Micro

Rain Radar and Ceilometer. We used COSMO-CLM² with an added aerosol-cycle module to test the cloud phase's sensitivity in response to varying prescribed INP concentrations. We tested two cases, one in austral summer, one in austral winter, and analysed the differences resulting from INP concentration changes for an area around the station and over the Southern Ocean within the selected domain. Our results show a strong influence of the INP concentration on the liquid water path in both regions, with higher concentrations reducing the amount of liquid water. Over the ocean, this effect is stronger during winter: During summer, a significant portion of water remains in liquid state regardless of INP concentration. Over the continent, this

effect is stronger during summer: Temperatures in winter frequently fall below $-37\,^{\circ}\mathrm{C}$, allowing homogeneous freezing. The largest increase of the liquid water fraction of total cloud hydrometeor mass is simulated over the ocean in winter, from 9.8 % in the highest tested INP concentration to 50.3 % in the lowest. The radiative effects caused by the INP concentration changes are small with less than $3\,\mathrm{W\,m}^{-2}$ difference in the averages between different concentrations.

## 1 Introduction

Microphysical properties of mixed-phase clouds are heavily influenced by ice nucleating particles (INPs; Kanji et al., 2017), which grow efficiently via depositional growth (Korolev, 2007; Morrison et al., 2012). In polar regions, and particularly in Antarctica, where INPs are very sparse compared to mid-latitude regions, only a few cloud droplets freeze at temperatures above about $-37\,^{\circ}\mathrm{C}$, the temperature at which water freezes homogeneously (Murray et al., 2010). Faster growth and larger



sedimentation velocities of the relatively few ice crystals, generate the often observed phase separation near cloud top. This results in a two-layer mixed-phase cloud (MPC), with a liquid layer above an ice layer, a common cloud type in the Arctic region that has also been observed in Antarctica (Gorodetskaya et al., 2015; Bromwich et al., 2012).

The cloud phase is known to have impacts on the radiative forcings exerted by the cloud (Van Tricht et al., 2016; Matus and L'Ecuyer, 2017; Hogan et al., 2003). In general, cloud radiative effects (CREs) can be summarised using the Eq. 1, where

$SW^{net}_{all-sky}$ denotes the total net downward shortwave (SW) radiation at the surface (i.e., $SW^{\downarrow}_{surface} - SW^{\uparrow}_{surface}$, so that a higher $SW^{net}_{all-sky}$ means more downwelling shortwave radiation reaching the surface without being reflected), $SW^{net}_{clear-sky}$ the net SW downward radiation without clouds, and their longwave (LW) equivalents being $LW^{net}_{all-sky}$ and $LW^{net}_{clear-sky}$:

$$CRE = CRE_{SW} + CRE_{LW} = SW^{net}_{all-sky} - SW^{net}_{clear-sky} + LW^{net}_{all-sky} - LW^{net}_{clear-sky} \qquad (1)$$

A cloud containing liquid water impacts CRE by decreasing $SW^{net}_{all-sky}$ compared to an ice-only cloud in the form of in-
creasing the albedo of the cloud and reflecting more sunlight back into space, and increasing $LW^{net}_{all-sky}$, i.e. increasing opacity and reflecting more radiation back to earth. As a result, through changing the cloud phase, INPs can have an influence beyond ice-water conversion. Because the connections between those effects are not yet very well understood, cloud feedback effects are currently the biggest source of uncertainty for modelled radiative forcings, not only in polar regions at present (Goosse et al., 2018) but also within global future climate scenarios (IPCC, 2023). Additionally, the cloud phase has a significant impact on
the rate of surface melt (Gilbert et al., 2020).

In most weather and climate models, there is a positive bias in the net SW radiation ($SW^{net}_{all-sky}$) over the Southern Ocean (Kay et al., 2016). This has been attributed to an underestimation of supercooled liquid water in clouds in those models. This problem is also affecting future scenarios, as with rising temperatures, the liquid water content of clouds is expected to be rising more quickly than the ice content (Chyhareva and Krakovska, 2022). In models that resolve this bias, it is often at the expense
of other modelling errors, such as an overall increase in total cloud water content (i.e., both liquid and ice) in the Met Office Unified Model (Brown et al., 2012; Van Weverberg et al., 2023, UM;). The model of the Consortium for Small-scale Modeling (COSMO) in climate mode (CLM), coupled to the Community Land Model (CLM), COSMO-CLM$^2$, simulates radiation in Antarctica with a mean absolute error between 7 and 20 $\mathrm{Wm}^{-2}$ for different radiation components ($SW^{\downarrow}$, $SW^{\uparrow}$, $LW^{\downarrow}$, and $LW^{\uparrow}$) and no significant SW biases, except for a bias in $LW^{\downarrow}$ during winter, which is on average -20 $\mathrm{Wm}^{-2}$ too low, once
again linked to an underrepresentation of liquid water in clouds (Souverijns et al., 2019). The aforementioned SW bias over the Southern Ocean does not affect continental Antarctica very much, thanks to the ice sheet's high albedo. Overall, this is comparable to other models: The community earth system model (CESM) has a 30 $\mathrm{Wm}^{-2}$ (warm) bias in $CRE_{SW}$ and a -10 $\mathrm{Wm}^{-2}$ (cold) bias in $CRE_{LW}$ over the Southern Ocean (Kay et al., 2012). In an ensemble mean of CMIP5 models, the SW bias over the Southern Ocean was found to be 20 $\mathrm{Wm}^{-2}$ (Hwang and Frierson, 2013). In West Antarctica, the ERA5 and
AMPS reanalyses were found to have a 14 $\mathrm{Wm}^{-2}$ and 21 $\mathrm{Wm}^{-2}$ LW bias respectively, with spikes of up to 50 $\mathrm{Wm}^{-2}$ when liquid, or mixed-phase clouds, were present (Silber et al., 2019). These biases are not exclusive to Antarctica either, as in the



northern polar region, a wide range of reanalyses (ERA5, ERA-Interim, CFSv2, MERRA2, JRA55, and ASRv2; Graham et al., 2019) show a negative bias in LW balance of 3 to 19 $\mathrm{Wm}^{-2}$.

There have been attempts to reduce the radiation biases through correcting the liquid water content: Supercooled liquid water clouds that were observed using the station's instruments at Dome C were modelled in two case studies using the regional climate model (RCM) ARPEGE-SH (Action de Recherche Petite Echelle Grande Echelle – Southern Hemisphere; Ricaud et al., 2020). By adding a liquid water partition function, they managed to remove the LW bias in one of their two case studies under stable atmospheric conditions, whereas in their second case study, which featured a capping inversion and was generally warmer, liquid water amounts were still too low and the radiation bias persisted. In ICON (icosahedral

nonhydrostatic weather and climate model), a bias in SW radiation balance was found to be caused by an underestimation of the cloud layer's thickness, liquid water content, and hydrometeor number concentration. Changing the cloud condensation nuclei (CCN) activation scheme reduced the bias in ICON, but did not fully resolve it (Kretzschmar et al., 2020). Another approach has been the implementation of macrophysical schemes for the Met Office UM (Van Weverberg et al., 2023), aimed at improving the representation of subgrid cloud structures, and while it was shown that these schemes have an influence,

they cannot fully resolve the issue, with liquid water contents remaining underestimated. Very few studies exist in which the influence of INPs as a potential source for liquid water is tested in a model. One notable exception to this is a study by Vignon et al. (2021), where different INP parametrisations are used in the WRF (Weather Research and Forecasting) model for a short case study in the austral summer, with results indicating a strong link between INP concentration and liquid cloud water content.

Even though most RCMs are optimized for mid-latitude performance, some adaptations for aiming to model the climate in Antarctica, as well as the Arctic, already exist. For COSMO-CLM[2], such adaptations have been done in a study (Souverijns et al., 2019) in which the atmospheric part of the model has been modified by reducing the modelled turbulence and thermal circulation, as well as the surface part by changing the snow properties. These changes lead to a good agreement for the modelled temperature, wind, and surface mass balance with the observations, although, as previously noted, biases in LW

radiation remain. This model is now also part of the Polar CORDEX (Coordinated Regional Climate Downscaling Experiment; Giorgi et al., 2009) suite. A different modification was added to the model to investigate the impact of ship exhausts on clouds in the Arctic (Possner et al., 2017). This modification added an aerosol scheme, which resolves CCN and INP concentrations explicitly and uses a two-moment hydrometeor scheme (Seifert and Beheng, 2006). Another notable development is that of PARASO, which adds an ocean model, NEMO (nucleus for european modelling of the ocean), a sea ice model, LIM

(Louvain-la-Neuve sea-ice model), and a continental ice model, f.ETISh (fast elementary thermomechanical ice sheet model), to COSMO-CLM[2] (Pelletier et al., 2022).

It seems likely that the underestimation of liquid water in Antarctica at least partially originates from the optimisation of climate models for mid-latitude regions. As such, this also concerns their freezing schemes. This often implicitly assumes that INP are distributed homogeneously around the world, even though it is known that their concentration is much lower over

Antarctica. As a result, models may underestimate the liquid water fraction in this region at temperatures below $0\,^\circ\mathrm{C}$. With



most models not resolving INP explicitly, there is also little knowledge as to what impact a change in their concentration would have.

In the here presented study, we do not aim to explain the current radiation biases in climate models, as that would also involve much tuning and error compensation in other parts of COSMO-CLM$^2$. We rather aim at improving the understanding of the role of INPs by testing the sensitivity of the cloud phase with respect to the INP concentration by prescribing different concentrations in an RCM capable of simulating INPs explicitly. Contrary to Vignon et al. (2021), who use different INP parametrisations, a module is available where INP and CCN concentrations are treated as prognostic variables. In this paper, we test if there is significant variation between INP concentrations that are relevant for Antarctica today, using unique INP measurements from PEA, indicating a need for a detailed simulation of INP in climate and weather models; and if there are significant changes to be expected when using INP concentrations measured in mid-latitude regions. The latter is relevant for climate and weather models, given their focus on these regions, but also, it might become relevant should INP concentrations in Antarctica change in the future: Twohy et al. (2021) suggest that a decrease in Antarctic Sea Ice and an increase of water temperatures in the southern ocean could result in an increase of INP concentrations in Antarctica. Finally, we can assess if the result that is best agreeing to our cloud observations also matches our INP observations.

## 2  Observations

The Belgian Princess Elisabeth Station Antarctica (PEA) is a zero-emission research base, located in Queen Maud Land, close to the Sør Rondane mountains, at 70° 57' S, 23° 20' E and 1390 m above mean sea level. It is inhabited during the Antarctic summer between November and February and is operated via remote access during the other months. It is close to the Antarctic plateau (50 km) and the Ragnhild coast (200 km), and is located in a relatively mild microclimate. The site is dominated by an easterly wind year-round (> 90 %) and air temperatures vary between −36 °C and −5 °C (Gorodetskaya et al., 2013; Pattyn et al., 2010). At PEA, an extensive weather and cloud observatory has been installed in 2009 (Gorodetskaya et al., 2015). This observatory consists of an automated weather station (AWS), a micro rain radar (MRR), and a ceilometer. While the AWS and MRR enable the detection of snowfall rates and properties like fall speed and temperature, the ceilometer detects cloud properties such as cloud height, and also facilitates cloud phase estimation (Guyot et al., 2022). Radio soundings by weather balloons delivered vertical profiles of temperature, humidity, pressure and wind. These soundings have been performed since season 2014/2015 during each austral summer up to now, except 2016/17.

In addition to the weather and cloud observations, ground-based INP measurements were taken in the 2020/21 and 2021/22 austral summers. These INP measurements were taken using 47 mm polytetrafluorethylene filters, which were set up in a shelter around 500 m north of PEA station. The filter holder was situated outside, on top of the shelter with a 15 cm long piece of flexible conductive tubing as inlet. Pump and flowmeter were inside the shelter. The sample volume flow was around 24 L min$^{-1}$ for the 2020/21 filters and between 15 and 25 L min$^{-1}$ for the 2021/22 filters. Temperature and pressure for calculation of ambient and standard conditions were taken from the AWS of PEA and from the flowmeter connected to the sampling line, respectively. Sample duration was around 10 days per filter. Each season, blank samples were taken. The subsequent




measurements were done in the same way as in Sze et al. (2022), using the two well-established off-line techniques LINA
(Leipzig ice nucleation array) and INDA (ice nucleation droplet array; Lacher et al., 2023). Our observations at PEA are
compared here with observations taken from literature in order to identify suitable INP concentrations to use for the sensitivity
experiments performed with COSMO-CLM[2].

The observations at PEA indicate substantial temporal variability in the concentrations, with concentrations varying from
$6 \times 10^{-6}$ to $5 \times 10^{-3}$ active INP per liter at an activation temperature of $-20\,^\circ$C. To simplify comparisons, we compared all
measured concentrations at a $-20\,^\circ$C reference temperature and converted measurements only available at other temperatures
using the parametrisation of DeMott et al. (2010) (see also Formula 2). Other recent INP measurements taken over the ocean
around Antarctica are slightly higher: Tatzelt et al. (2022) and McCluskey et al. (2018) measured similar concentrations over the
Southern Ocean at $3 \times 10^{-3}\,\mathrm{L}^{-1}$ to $3 \times 10^{-2}\,\mathrm{L}^{-1}$ and $3.8 \times 10^{-4}\,\mathrm{L}^{-1}$ to $4.6 \times 10^{-3}\,\mathrm{L}^{-1}$ respectively, but do not reach the extreme
low values we observed at PEA. Older observations, such as the ones by Bigg and Hopwood (1963) and Saxena and Weintraub
(1988), sometimes report much higher numbers with peaks of up to $13\,\mathrm{L}^{-1}$ (Bigg and Hopwood, 1963). Given the the large
number of more recent observations with much lower results, the validity of this exceptionally high result may be questioned. It
is especially remarkable when compared to recent measurements in other regions which are known to experience higher aerosol
concentrations: Chen et al. (2018) measured INP concentrations of up to $2\,\mathrm{L}^{-1}$ in Beijing; Petters and Wright (2015) report a
similar amount in North Carolina, with lower bounds of $3 \times 10^{-1}\,\mathrm{L}^{-1}$ in Beijing and $3 \times 10^{-2}\,\mathrm{L}^{-1}$ in North Carolina. Peak
concentrations might be higher than the observations presented here, as filter measurements are typically collecting INPs over
the course of several hours. Compared to the newer Antarctic measurements, both the observations by (Bigg and Hopwood,
1963), and the more recent observations elsewhere in the world deliver high results nonetheless. Even though mid-latitude
oceans (e.g. Welti et al., 2020, $5 \times 10^{-3}\,\mathrm{L}^{-1}$ to $1 \times 10^{-1}\,\mathrm{L}^{-1}$; Raman et al., 2022, $1 \times 10^{-2}\,\mathrm{L}^{-1}$ to $1 \times 10^{0}\,\mathrm{L}^{-1}$) observe
slightly lower values than those in Chen et al. (2018) and Petters and Wright (2015), it can be concluded that overall, INP
number concentrations in Antarctica are exceptionally low. This is in line with Raman et al. (2022), who found that high INP
concentrations at Macquarie Island are correlated to organic matter and dust emission events occurring in nearby New Zealand
and favorable conditions for phytoplankton growth, both of which seem unlikely to frequently happen in such a proximity
to PEA that it would have a significant impact on INP concentrations. Twohy et al. (2021) however suggest that at least the
phytoplankton activity might increase in future climate scenarios. Table 1 provides an overview of the INP measurements taken
into account when selecting the prescribed concentrations.

The combination of these measurements make PEA an ideal site for investigating aerosol-cloud interactions, as nowhere
else on the continent, simultaneous gound-based radar, lidar, and INP measurements are available. In addition to that, the zero-
emission approach of the station allows us to investigate atmosphere and clouds without disturbances by emissions from the
station. We assume that the concentration of INPs will have a significant impact on the cloud phase, with a lower concentration
limiting the amount of ice production, and in turn, we expect a stronger CRE, both for SW radiation, decreasing $SW_{all-sky}^{net}$
and LW radiation, increasing $LW_{all-sky}^{net}$.





| Reference | Region | Method | Active INP [$L^{-1}$] |
|---|---|---|---|
| Own measurements | PEA | LINA; INDA (filters) | $6 \times 10^{-6}$ to $5 \times 10^{-3}$ |
| Tatzelt et al. (2022) | Southern Ocean | DIGITEL low volume sampler (filters) LINA; INDA (filters) | $3 \times 10^{-3}$ to $3 \times 10^{-2}$ |
| McCluskey et al. (2018) | South of Australia, maritime | Continuous flow diffusion chamber Ice spectrometer (filters) | $3.8 \times 10^{-4}$ to $4.6 \times 10^{-3}$ |
| Bigg and Hopwood (1963) | McMurdo | Mixing cold chamber | $5 \times 10^{-1}$ to $13 \times 10^{0}$ |
| Raman et al. (2022) | Macquarie Island | Filters | $1 \times 10^{-2}$ to $1 \times 10^{0}$ |
| Welti et al. (2020) | Northern Temperate Zone, maritime | various | $5 \times 10^{-3}$ to $1 \times 10^{-1}$ |
| Chen et al. (2018) | Beijing, China | LINA; INDA (filters) | $3 \times 10^{-1}$ to $2 \times 10^{0}$ |
| Petters and Wright (2015) | Raleigh, North Carolina, USA | drop-freezing assay (Glass dishes) | $3 \times 10^{-2}$ to $2 \times 10^{0}$ |

**Table 1.** Overview of different INP measurements taken into account for scenario selection. All measured concentrations were converted to a $-20\,°C$ activation temperature using the parametrisation of DeMott et al. (2010). The second horizontal line separates Antarctic and Southern Ocean measurements from measurements in other regions.

# 3 Methodology

## 3.1 Model Description

In this study, we deploy COSMO-CLM$^2$ version 5.0, using the combined modifications done by Souverijns et al. (2019) and
Possner et al. (2017). COSMO-CLM$^2$ consists of two main components: The COSMO regional atmosphere model in climate mode (COSMO-CLM; Steger and Bucchignani, 2020), which is maintained by the Climate limited-area modelling community (CLM-Community), and the community land model (CLM, Oleson et al., 2013), which is the land component of the community earth system model (CESM). Those two models are coupled using the OASIS Model Coupling Toolkit (OASIS3-MCT; Will et al., 2017; Craig et al., 2017). The changes made by Souverijns et al. (2019) improve the representation of the Antarctic
climate in the model through optimisations and reimplementations of surface snow and ice sheet parametrisations, changing the roughness length of snow for a correct representation of katabatic winds, and changing the settings of the turbulent kinetic energy scheme to account for the more stable atmosphere over the Antarctic Ice Sheet.

The aerosol and ice nucleation module (Possner et al., 2017) improves the parametrisation of cloud microphysics by resolving CCN and INPs explicitly, based on the parametrisation described by Solomon et al. (2015). Hydrometeors are simulated
according to the two-moment scheme by Seifert and Beheng (2006). This module adds 16 different INP concentrations as a variable each, with the different bins corresponding to different activation temperatures. The first variable stores the concentration of available INPs that activate at or above $258.15\,K$ ($-15\,°C$). The remaining 15 variables store the INPs activating at lower temperatures with each activation temperature being $1.3\,K$ colder than the previous, i.e., the second bin contains the concentration of INPs activating between $258.15\,K$ and $256.85\,K$. This places the lowest temperature at $238.65\,K$ ($-34.5\,°C$),
close to the temperature at which homogeneous freezing starts to occur. For each of the 16 concentration bins, half of the INPs





activate per simulation timestep if the temperature is below the bin's temperature, converting an equal amount of supercooled liquid water particles (if available) into ice particles. The used INPs are then depleted, reducing their concentration, but can be reintroduced by sublimation of snow or ice particles. The module also accounts for secondary ice production (SIP), but only in the mode of Rime Splintering (Hallett and Mossop, 1974).

For the initial and boundary conditions, we prescribe one INP concentration per integration, given as the concentration of INPs activating at or above $253.15\,\mathrm{K}$ ($-20\,^\circ\mathrm{C}$). The individual number concentrations $N_{INP}(T)$ for the different activation temperatures $T$ in Kelvin are then derived using Formula 2 (DeMott et al., 2010; Solomon et al., 2015), where $F$ is a scaling factor, chosen so that $N_{INP}(253.15)$ results in the prescribed concentration.

$$N_{INP}(T) = F^{1.25} exp\left(0.46\left(273.16 - T\right) - 11.6\right)) \tag{2}$$

In Eq. 2, $N_{INP}(T)$ describes the total number of INPs activating at or above a given temperature $T$ in Kelvin. For each activation temperature bin, except the first one with the highest temperature, the concentrations should however correspond to the INPs activating between the bin's activation temperature and the activation temperature of the previous bin. Therefore, the actual prescribed initial concentration will be the difference between the results of Formula 2 for the two temperatures, i.e., for the second bin, the prescribed concentration would be $N_{INP}(256.85) - N_{INP}(258.15)$.

As the amount of aerosol particles is subject to significant spatial and temporal variability (Raman et al., 2022; McFarquhar et al., 2020), the prescription of the same concentrations on all pressure levels at all times on the model boundaries is not a realistic assumption, even when allowing variation within the domain. However, prescribing different concentrations allows us to examine the potential effects that those different concentrations may have. Comparing the results of a model integration where we prescribe an INP concentration on the low end of the observed range in Antarctica to those of an integration where

we prescribe a concentration on the high end, we tested the model response to those different aerosol settings, and analysed the impact of aerosol concentrations compared to other variables. Furthermore, by comparing the results achieved with a low INP concentration to results achieved with mid-latitude concentrations, which are unrealistically high for the region, we examine how big the impact on clouds and precipitation is that these lower concentrations have. Finally, all of this output is compared to cloud observations taken at PEA to verify that cloud properties unrelated to INPs are well-represented and to get an idea of

what INP concentrations give the most realistic results when used with our model for the region.

## 3.2    Model Setup

Our simulation domain has a size of 400 by 400 grid points with a resolution of $0.025^\circ$ and is centered around PEA. Vertically, the grid consists of 40 levels, 18 of which are within 3km of the surface at sea level. For our boundary and initial conditions, we are using 3-hourly ERA5 data, and produce hourly output for the analysed variables, while the simulation timestep is 20 s.

We also did preliminary tests using a smaller domain with a size of 192 by 175 grid points in a nested configuration with the model output of Souverijns et al. (2019) as boundary conditions, but found that the clouds are much better represented in terms of height, timing, and structure when using the larger domain.





| Abbreviation | Name | INP concentration | Reference region |
|---|---|---|---|
| VL | very low | $1 \times 10^{-5}\,\text{L}^{-1}$ | low-end Antarctica |
| L | low | $5 \times 10^{-3}\,\text{L}^{-1}$ | high-end Antarctica |
| M | medium | $5 \times 10^{-2}\,\text{L}^{-1}$ | maritime; high-end Antarctica with SIP augmentation |
| H | high | $2 \times 10^{-1}\,\text{L}^{-1}$ | mid-latitude continental |
| VH | very high | $2\,\text{L}^{-1}$ | highest globally |

**Table 2.** Overview of prescribed INP settings. All concentratrions refer to the $-20\,°\text{C}$ temperature bin. L and H concentrations were not used for the winter period.

Based on the observations, we selected the following five INP concentration settings: First, we prescribe INP concentrations close to the lower end of the observed range in Antarctica at $1 \times 10^{-5}\,\text{L}^{-1}$ (at $-20\,°\text{C}$), named "very low" (VL, see Table 2). Second, we use a concentration close to the upper end of recently observed concentrations in Antarctica at $5 \times 10^{-3}\,\text{L}^{-1}$, named "low" (L). Third, we prescribe $5 \times 10^{-2}\,\text{L}^{-1}$, which we consider a realistic value for continental INP concentrations at more remote locations excluding Antarctica and maritime conditions, named "medium" (M). This medium concentration also serves as an augmentation for the SIP modes not captured by our setup (all except rime splintering), as it was found that these modes can increase ice crystal number concentrations (ICNC) by a factor of 10 (Sotiropoulou et al., 2020). It must be noted that this is not a very accurate assumption, as SIPs are only active very locally with great spatial variance (Georgakaki et al., 2022), but due to the scale of the model, the error caused by this is likely small. Fourth, we prescribe $2 \times 10^{-1}\,\text{L}^{-1}$ as the first control run with continental concentrations, named "high" (H); And fifth, $2\,\text{L}^{-1}$ is used as the maximum prescribed concentration, named "very high" (VH), which corresponds to the maximum observed concentrations. It should also be noted that this is in the range observed by (Bigg and Hopwood, 1963), which, while we do not consider it to be an accurate measurement anymore, showcases the large variability of possible concentrations. Table 2 gives a summary of the settings used. In addition to INP concentrations, concentrations of CCN can also be prescribed in the aerosol module. However, we performed initial tests, varying the prescribed CCN concentrations from $10\,\text{cm}^{-3}$ to $1300\,\text{cm}^{-3}$, corresponding to the measured range at PEA (Herenz et al., 2019). These tests showed, agreeing with previous findings (Solomon et al., 2018), a negligible impact of CCN concentrations on cloud phase, which is why the impact of CCN concentrations was not investigated further. Thus, in all of our simulations, we used the low-end CCN concentration of $10\,\text{cm}^{-3}$.

We selected the periods to simulate based on two important factors: First, observations of both the MRR and the ceilometer should be available to be able to control the accuracy of the model output. Second, there should be a variety of observed cloud features to test the model under different conditions. Furthermore, one of the runs should be in the summer, and the other during the winter to see if the differences in temperature and radiation between the seasons have an impact on the results.

The first simulated period spans 40 days in the summer from the 10 January 2012, to the 19 February 2012. This period was selected because it is the period with the most variation in cloud types in a given amount of time recorded by the ceilometer, and because there is already a study identifying the different cloud types (Gorodetskaya et al., 2015): From the 6 February to the 7 February, the ceilometer registered a very optically thick layer, leading to quick extinction of the lidar signal, indicating



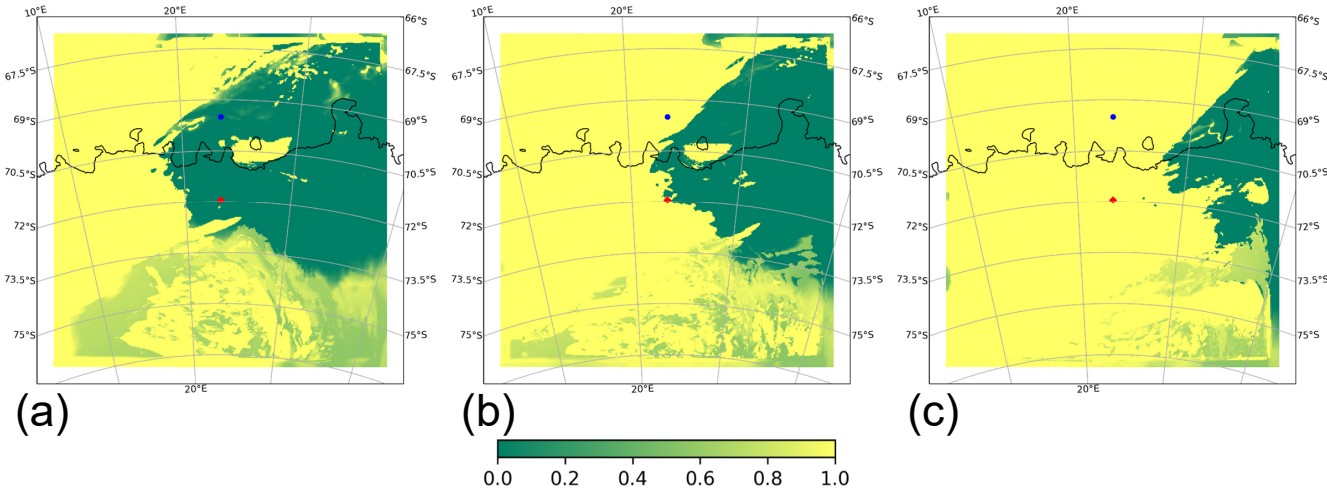

**Figure 1.** Low cloud cover fraction over the domain at 8 February 2012, 18:00 (a); 9 February 2012, 00:00 (b); and 9 February 2012, 06:00 (c). The red diamond denotes the location of the Princess Elisabeth station (PEA), the blue dot marks the location over the Southern Ocean we will analyse in more detail further on.

the occurrence of a liquid-containing cloud. Shortly after that, a frontal system passed over the station from the 8 to 11 of February, shown in Fig. 1, bringing snowfall to the station, as registered by the MRR, and causing the ceilometer signal to extinguish at a low level. There was, however, a short gap in the precipitation, where low-level mixed-phase clouds become visible (see MRR and ceilometer and MRR data in Fig. 2). This frontal system was quickly followed by a second, weaker one, passing the station from the 12 to 14 February, consisting of mostly ice clouds.

In the second period that was simulated, we looked at the Antarctic winter between the 20 July and 15 August 2022. Data availability for the cloud observatory is limited in winter periods, as the lack of sunlight and low temperatures limit power supply and the operationability of the instruments, so only recently, we were able to observe a full winter with all instruments. In the given time period, the observatory registered three major events: Firstly, between 25 and 27 July, intense snowfall can be seen on both the Ceilometer and the MRR; Secondly, a series of non-precipitating clouds passed over the station between 3 and 7 August, with no clear indications of liquid water; Thirdly, a similar cloud series passed over the station between 10 and 15 August. An overview for the MRR and Ceilometer measurements for the relevant periods can be found in Fig. A2.

## 4 Results

In the simulation, we found a connection between INP concentration and cloud water contents. The following figures are a representative selection to highlight our most important findings regarding that relationship.

In the summer period, the temperatures are sufficiently high, such that a limited amount of INPs results in supercooled liquid water persisting in some of the clouds, although the total amount of liquid water remains limited to a few spots. This





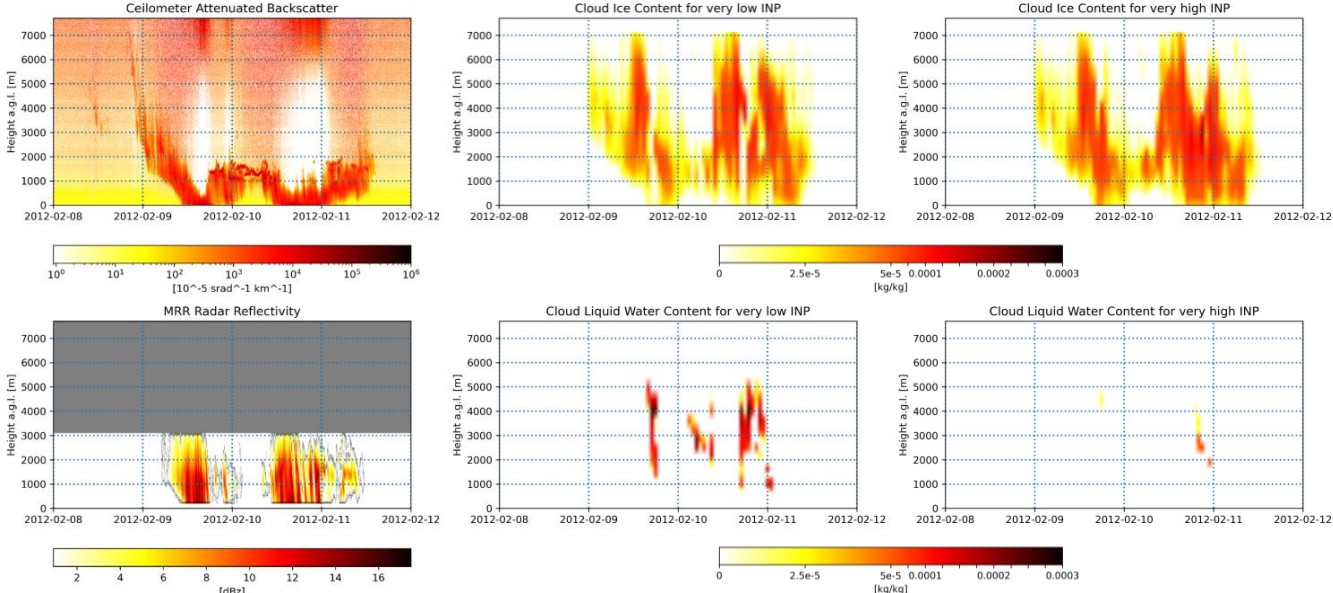

**Figure 2.** Cloud liquid water (bottom row) and ice (top row) for the VL (very low INP concentration, $1 \times 10^{-5}\,\mathrm{L}^{-1}$; middle column) and VH (very high INP concentration, $2\,\mathrm{L}^{-1}$; right column) settings, compared to the ceilometer (top left) and MRR (bottom left) measurements, in the time period 8 August 2012 to 12 August 2012.

behaviour can be seen in Fig. 2, where a significant portion of hydrometeors in the frontal cloud remain in a liquid state when using the VL setting, as opposed to the VH setting. Furthermore, the simulated cloud matches the observations in timing and cloud height well. Most of the liquid water simulated under VL conditions does however not form a thin, consistent layer, as expected and observed by the ceilometer around midnight on the 10 February between 1 km and 2 km height. Instead, the

liquid water reaches much higher altitudes of up to 5 km, in the area that could not be observed by the ceilometer due to signal extinction. This does not change significantly when looking at other areas over the continent within the domain. Consistent with the inability of the model to simulate a persistent layer of supercooled liquid water, the stratocumulus cloud observed by the ceilometer between 6 and 8 February, consisting of mostly liquid water, is not present in the model output at all (not shown).

In the winter period, the influence of the INP concentration has a much smaller effect, as shown in Fig. 3. The ceilometer data shows a cloud similar to the one seen in Fig. 3, but without the interruption in precipitation, and without a clear liquid layer. The MRR observations show that there is near-constant light precipitation during the depicted period. Again, all concentration variants capture the observed clouds well, with ice droplets reaching down to the surface between the early morning of the 25th and the late evening of 26 July 2022. Liquid water is, however, almost completely absent, except for a few patches in the

VL concentration at a height between 1000 and 2000 m. With the modelled temperature being at or slightly above 250 K at





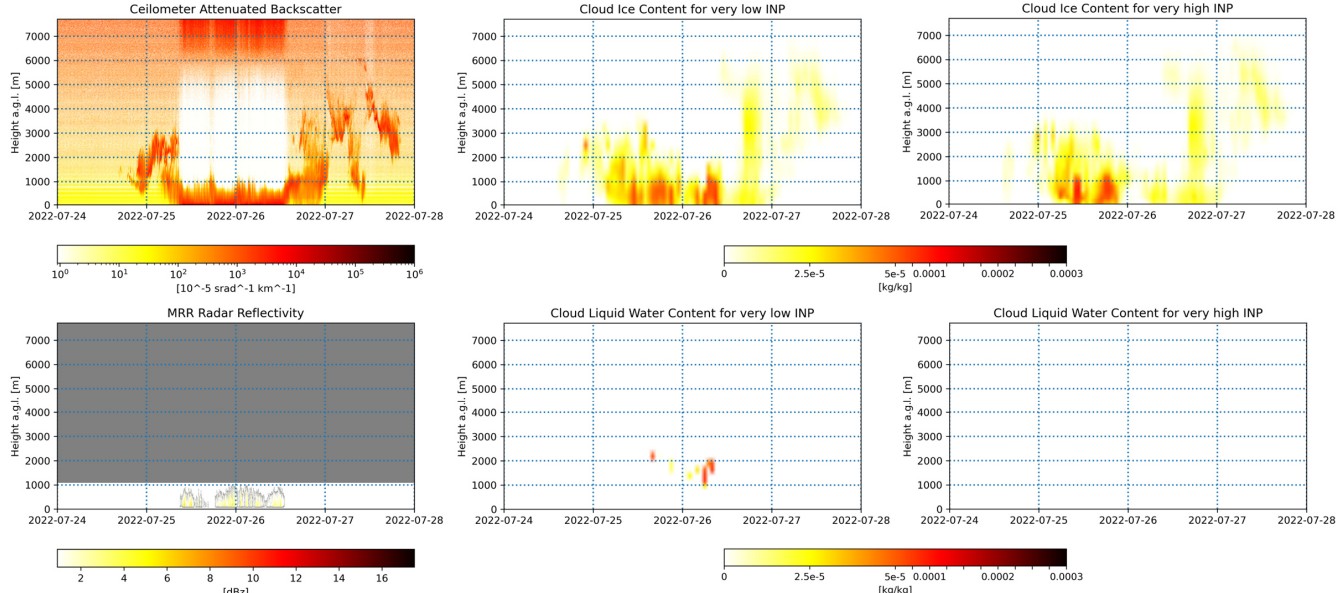

**Figure 3.** Cloud liquid water (bottom row) and ice (top row) for the VL (very low INP concentration, $1 \times 10^{-5}\,\mathrm{L}^{-1}$ at $-20\,^{\circ}\mathrm{C}$; middle column) and VH (very high INP concentration, $2\,\mathrm{L}^{-1}$ at $-20\,^{\circ}\mathrm{C}$; right column) settings, compared to the ceilometer (top left) and MRR (bottom left) measurements, in the time period 24 July 2022 to 28 July 2022. The MRR was set to only measure up to a height of 1 km at an increased resolution for this time period.

the surface (Fig. A1) during the period and decreasing with height, homogeneous nucleation offers a likely explanation for this reduced sensitivity.

The average amount of cloud liquid water varies significantly between the different concentrations, from 0.4 $\mathrm{gm}^{-2}$ at the VH concentration to 3.9 $\mathrm{gm}^{-2}$ at the VL concentration in the summer. Meanwhile, the change in cloud ice content at PEA across the different concentrations is small. Figure 4a shows that the absolute change in cloud ice content has a similar order of magnitude in absolute numbers, but in relation to the total content, this difference is much smaller (26.9 $\mathrm{gm}^{-2}$ in VL and 28.8 $\mathrm{gm}^{-2}$ in VH). In other words, the liquid mass fraction of hydrometeors ($\frac{TQC}{TQI+TQC}$) at PEA in the summer period increased to 12.6 % in VL, from 1.3 % in VH. When looking at the winter period (Fig. 4b), the influence of INP concentrations on cloud properties at the station is drastically reduced (VL: liquid mass fraction of hydrometeors 4.0 %; VH: 0.1 %), which is in line with our expectation that extremely low temperatures allow widespread homogeneous freezing and the behaviour seen in Fig. 3. All of these values are averaged over a 21 by 21 grid cell area centered at PEA.

The radiative effects caused by these changes in cloud phase are small. Figure 5a shows that in general, the median and mean cloud radiative effects generally stay between 50 $\mathrm{Wm}^{-2}$ and 60 $\mathrm{Wm}^{-2}$ for the summer period, with the extremes being slightly lower in the VH setting, with no clear trends connected to INP concentrations, and only the M vs. VH concentrations show a statistically significant difference in the paired t-test (Table 3). When the total CRE is split up into a shortwave and longwave part (Fig. 5b and c), the means of the shortwave CRE are decreasing towards a lower INP concentration, indicating





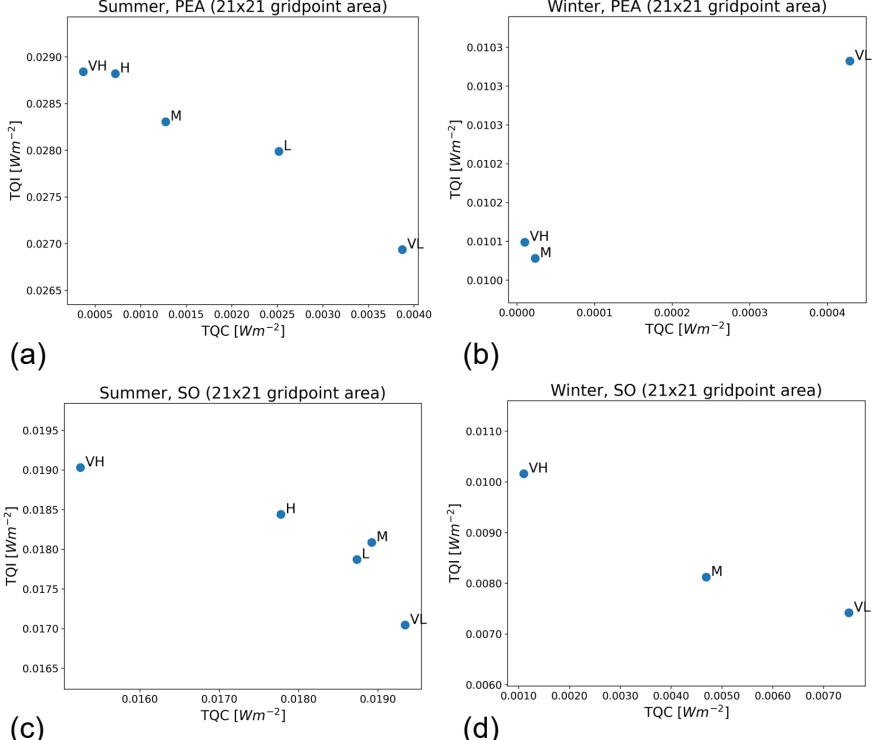

**Figure 4.** Average amounts of vertically integrated liquid water and ice under different INP concentrations over the summer period (10 January 2012 to 18 February 2012, left) and the winter period (20 July 2022 to 15 August 2022, right) at a 21 by 21 gridpoint box around PEA (Princess Elisabeth Station Antarctica, top) and around 69.5° S, 23.35° E in the Southern Ocean (SO, bottom). TQI = vertically integrated cloud ice, TQC = vertically integrated cloud water. VL = very low INP concentration, $1 \times 10^{-5}\,\mathrm{L}^{-1}$; L = low INP concentration, $5 \times 10^{-3}\,\mathrm{L}^{-1}$; M = medium INP concentration, $5 \times 10^{-2}\,\mathrm{L}^{-1}$; H = high INP concentration, $2 \times 10^{-1}\,\mathrm{L}^{-1}$, VH = very high INP concentration, $2\,\mathrm{L}^{-1}$; all INP concentrations at $-20\,°\mathrm{C}$.

that the higher liquid share is more optically thick and therefore reflects a higher portion of sunlight back to space. This is offset by the trend of the longwave CRE, which is increasing towards a lower INP concentration, indicating that the higher liquid portion also reflects more radiation back to the ground. If we, however, only look at the time steps with significant liquid water

present (Fig. 5d), we can see that in the VL and L concentrations, the mean CRE is significantly lower than in the cases with higher INP concentrations (VL: 65.4 $\mathrm{Wm}^{-2}$, L: 67.2 $\mathrm{Wm}^{-2}$; M: 68.8 $\mathrm{Wm}^{-2}$, H: 69.4 $\mathrm{Wm}^{-2}$, VH: 69.7 $\mathrm{Wm}^{-2}$), indicating that for these thicker clouds, the increased shortwave reflection outweighs the longwave reflection. The sample size, however, gets rather small as there are only 11 timesteps with sufficient liquid water available to meet our criteria when averaging over the 21 by 21 gridpoint area.

We also analysed the phase of hydrometeors found further north over the Southern Ocean at the grid cells around 69.5° S, 23.35° E. Over the ocean, the average air temperature is warmer, and as such, liquid water is more common, even when







**Figure 5.** Different CRE (cloud radiative effects) statistics for the summer (a-f) and winter (g) period, averaged over a 21x21 area around the grid cell of PEA. TQI = vertically integrated cloud ice, TQC = vertically integrated cloud water. The blue markers indicate individual timesteps, the red solid line the median, and the black dashed line indicated the mean. Sample sizes for the subfigures: (a, b, c): 131, (d, e, f): 11, (g): 61. VL = very low INP concentration, $1 \times 10^{-5}\,\mathrm{L}^{-1}$; L = low INP concentration, $5 \times 10^{-3}\,\mathrm{L}^{-1}$; M = medium INP concentration, $5 \times 10^{-2}\,\mathrm{L}^{-1}$; H = high INP concentration, $2 \times 10^{-1}\,\mathrm{L}^{-1}$, VH = very high INP concentration, $2\,\mathrm{L}^{-1}$; all INP concentrations at $-20\,°\mathrm{C}$.



|  | VL | L | M | H | VH |
|---|---|---|---|---|---|
| VL | - |  | d,g | d | d,g |
| L |  | - | d | d | d |
| M | d,g | d | - |  | a,d |
| H | d | d |  | - |  |
| VH | d,g | d | a,d |  | - |

**Table 3.** Samples with a significant (p < 0.05) difference in total CRE (cloud radiative effects) between two INP concentrations averaged over a 21x21 grid cell area around the grid cell of the Princess Elisabeth station, tested by the paired t-test. Letters a, d, and g refer to the selection critera and periods as shown in the subfigures of Fig. 5 (a = summer period, only timesteps with $TQI + TQC > 5 \times 10^{-5} \mathrm{g\,m^{-2}}$ in at least one of the INP concentrations; d = summer period, only timesteps with $TQC > 1.5 \times 10^{-4} \mathrm{g\,m^{-2}}$; g = winter period, only timesteps with $TQI + TQC > 5 \times 10^{-5} \mathrm{g\,m^{-2}}$). TQI = vertically integrated cloud ice, TQC = vertically integrated cloud water. VL = very low INP concentration, $1 \times 10^{-5} \mathrm{L^{-1}}$; L = low INP concentration, $5 \times 10^{-3} \mathrm{L^{-1}}$; M = medium INP concentration, $5 \times 10^{-2} \mathrm{L^{-1}}$; H = high INP concentration, $2 \times 10^{-1} \mathrm{L^{-1}}$, VH = very high INP concentration, $2\mathrm{L^{-1}}$; all INP concentrations at $-20\,°\mathrm{C}$.

prescribing a higher amount of INPs. Changing the INP concentration still has an impact on the cloud phase (see Fig. 4c), with the average amount of liquid water – averaged over a 21 by 21 grid cell area around the central point – increasing from 15.2 $\mathrm{gm^{-2}}$ in the VH to 19.3 $\mathrm{gm^{-2}}$ in the VL concentration for the summer period, while the cloud ice content decreases from 19.0

$\mathrm{gm^{-2}}$ to 17.0 $\mathrm{gm^{-2}}$. However, while this change is noticeable, it does not have an impact on the general structure of the cloud like it has on the clouds at PEA in Fig. 2, and the liquid mass fraction of hydrometeors changes only from 44.5 % in VH to 53.2 % in VL. During the winter period in the Southern Ocean, the INP concentration has a much larger impact, both compared to the summer period at the same location and the winter period at PEA, with the amount of cloud liquid water ranging from 1 $\mathrm{gm^{-2}}$ at VH to 7.5 $\mathrm{gm^{-2}}$ at VL concentrations for liquid water mass fractions of 9.8 % (VH) and 50.3 % (VL; see Fig. 4d).

Overall, the radiative effects of clouds are stronger over the Southern Ocean than over the continent. As shown in Fig. 6a, when analysing the summer period's timesteps with liquid or ice hydrometeors above 0.05 $\mathrm{gm^{-2}}$ available, the median CREs are all around the 0 $\mathrm{Wm^{-2}}$ mark, with a maximum of 5.4 $\mathrm{Wm^{-2}}$ for the M concentration, but with the total CRE going down to extremes of -500 $\mathrm{Wm^{-2}}$ in some timesteps. When applying the more constraining condition that only counts timesteps whith at least 0.15 $\mathrm{gm^{-2}}$ of liquid water available, it can be seen that the radiative effects of these clouds are much stronger, placing

the medians between -45 $\mathrm{Wm^{-2}}$ and -50 $\mathrm{Wm^{-2}}$ (VL: -46.3 $\mathrm{Wm^{-2}}$, L: -45.5 $\mathrm{Wm^{-2}}$, M: -47.9 $\mathrm{Wm^{-2}}$, H: -45.9 $\mathrm{Wm^{-2}}$, VH: -48.0 $\mathrm{Wm^{-2}}$), highlighting the stronger radiative effects of liquid clouds. However, during the entire summer period, there is no clear difference between the different concentrations. Only when comparing the VL with the VH INP concentration for the stricter condition of 0.15 $\mathrm{gm^{-2}}$ of liquid water, the difference per timestep is statistically significant (Table 4).

During winter, the median CRE shows an increase when comparing the VL concentration to VH and M (Fig. 6g: VL: 81.6

$\mathrm{Wm^{-2}}$, M: 79.8 $\mathrm{Wm^{-2}}$, VH: 79.4 $\mathrm{Wm^{-2}}$) over the ocean, using the more relaxed condition of at least 0.05 $\mathrm{gm^{-2}}$ of water and ice. This is in line with our findings from Fig. 5c and f, where we found an increase in the longwave part of CRE for liquid-containing clouds: During the polar winter, the clouds cannot reflect any sunlight, thus putting the shortwave CRE to 0

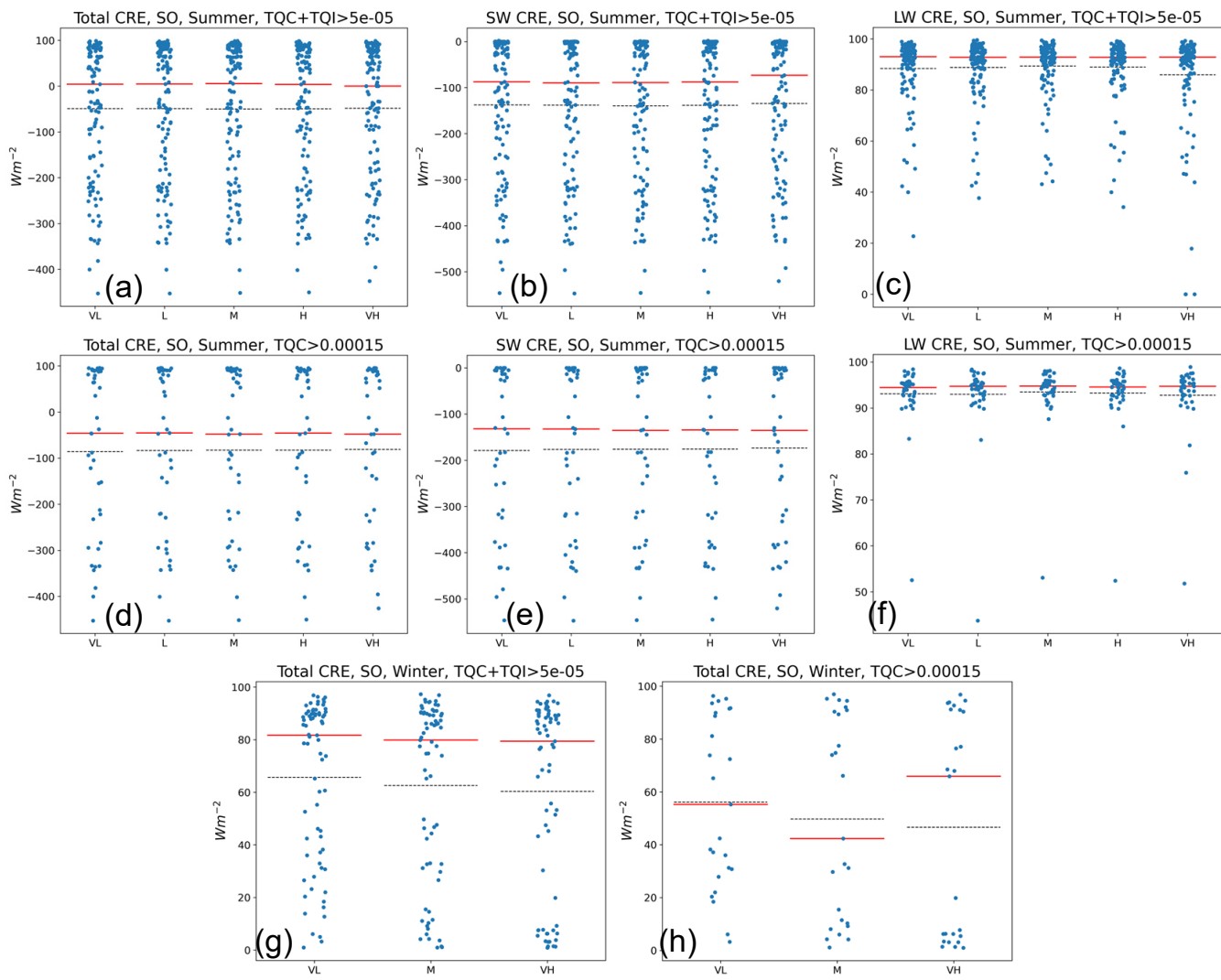

**Figure 6.** Different CRE (cloud radiative effects) statistics for the summer (a-f) and winter (g, h) period, averaged over a 21x21 area around 69.5°S, 23.35°E in the southern ocean. TQI = vertically integrated cloud ice, TQC = vertically integrated cloud water. The blue markers indicate individual timesteps, the red solid line the median, and the black dashed line indicated the mean. Sample sizes for the subfigures: (a, b, c): 121, (d, e, f): 41, (g): 71, (h): 25. VL = very low INP concentration, $1 \times 10^{-5}\,\mathrm{L}^{-1}$; L = low INP concentration, $5 \times 10^{-3}\,\mathrm{L}^{-1}$; M = medium INP concentration, $5 \times 10^{-2}\,\mathrm{L}^{-1}$; H = high INP concentration, $2 \times 10^{-1}\,\mathrm{L}^{-1}$, VH = very high INP concentration, $2\,\mathrm{L}^{-1}$; all INP concentrations at $-20\,°\mathrm{C}$.



|     | VL     | L | M   | H | VH    |
| --- | ---    | --- | --- | --- | ---   |
| VL  | -      |   | g,h |   | d,g,h |
| L   |        | - |     |   |       |
| M   | g,h    |   | -   |   | g     |
| H   |        |   |     | - |       |
| VH  | d,g,h  |   | g   |   | -     |

**Table 4.** Samples with a significant (p < 0.05) difference in total CRE (cloud radiative effects) between two INP concentrations averaged over a 21x21 grid cell area around 69.5°S, 23.35°E in the southern ocean, tested by the paired t-test. Letters a, d, g, and h refer to the selection critera and periods as shown in the subfigures of Fig. 6 (a = summer period, only timesteps with $TQI + TQC > 5 \times 10^{-5} \mathrm{g\,m}^{-2}$ in at least one of the INP concentrations; d = summer period, only timesteps with $TQC > 1.5 \times 10^{-4} \mathrm{g\,m}^{-2}$; g = winter period, only timesteps with $TQI + TQC > 5 \times 10^{-5} \mathrm{g\,m}^{-2}$; h = winter period, only timesteps with $TQC > 1.5 \times 10^{-4} \mathrm{g\,m}^{-2}$). TQI = vertically integrated cloud ice, TQC = vertically integrated cloud water. VL = very low INP concentration, $1 \times 10^{-5} \mathrm{L}^{-1}$; L = low INP concentration, $5 \times 10^{-3} \mathrm{L}^{-1}$; M = medium INP concentration, $5 \times 10^{-2} \mathrm{L}^{-1}$; H = high INP concentration, $2 \times 10^{-1} \mathrm{L}^{-1}$, VH = very high INP concentration, $2 \mathrm{L}^{-1}$; all INP concentrations at $-20\,°C$.

(this is also why only total CREs are shown for the winter period). As Fig. 6g shows, the increase is much more pronounced in the region close to 0 $\mathrm{Wm}^{-2}$, where a lot of the samples of the VH concentration are clustered close to 0 $\mathrm{Wm}^{-2}$, while that cluster spreads out to 20-40 $\mathrm{Wm}^{-2}$ much more for the M and VL concentrations. This indicates that it is the less optcally dense clouds whose CRE is enhanced the most when limiting available INPs. When limiting the timesteps to 0.15 $\mathrm{gm}^{-2}$ of liquid cloud water (Fig. 6h), this effect gets even stronger: there is no longer a clear signal in the median, likely as it falls in a region with very few samples, but, while the area above the median line has a similar distribution, the area closer to 0 $\mathrm{Wm}^{-2}$ is very spread out in VL and very clustered close to 0 $\mathrm{Wm}^{-2}$ in VH, with M being in between. This is also reflected by the mean. With only 25 timesteps available when applying this limitation, the sample size is very small, but still sufficient to find a significant (p < 0.5) difference of the VL to the other concentrations (Table 4).

## 5 Discussion

In summer, a clear relation was found between the simulated concentration of INPs and the presence of liquid water in the clouds at PEA. The higher amounts of liquid water in the simulations with a limited INP concentration (VL and L) agree better with the ceilometer and MRR observations. While this overall improves the representation of the cloud phase, the model still fails to accurately represent the long-lived, dense liquid and mixed-phase layers observed at the station. This effect is not caused by restricting our view to a single grid cell, as it can also be seen when averaging the liquid water content over an area of 10 grid cells to all sides of PEA. The radiative effects caused by the added liquid are noticable, but in general, do not affect the overall radiative balance of the model significantly. It should be noted, for both PEA and the Southern Ocean case, that stratocumulus clouds, which the model was unable to represent, were very likely to have supercooled liquid water (Gorodetskaya et al., 2015; Twohy et al., 2021), which might result in an underestimation of cloud liquid water content, and therefore, the radiative effects.




The differences that we found generally support the findings of (Ricaud et al., 2024), who estimate that over the Antarctic Plateau, supercooled liquid water only has a weak radiative effect of up to $7\,\mathrm{Wm^{-2}}$, as opposed to up to $40\,\mathrm{Wm^{-2}}$ over the Antarctic Peninsula, and who found a clear correlation between cloud liquid water content and temperature.

Over the Southern Ocean, the relation between INP concentrations and liquid water presence was also found, but even in the highest INP setting, a significant amount of liquid water remains. This may be explained by the higher overall temperature, as even with temperatures below $0\,^\circ$C, not all INPs activate immediately. In fact, the first INP temperature bin of our model activates at $-15\,^\circ$C, so at higher temperatures, ice production is limited to secondary processes. However, the cloud top temperatures we observed in the model during that time period were around or below $-15\,^\circ$C (not shown). As the variability of
the liquid mass fraction of hydrometeors is relatively small, unlike at PEA, it seems unlikely that in the summer months, INP concentrations have noticable impacts on the cloud phase over the ocean in our model.

During the winter period, the relationship between INP concentrations and liquid water concentration changes. On the one hand, at PEA, only small amounts of liquid water remain in the clouds. As temperatures over the inland regions of Antarctica commonly reach below the threshold of $-37\,^\circ$C required for homogeneous nucleation in our model, even at the surface level,
this is not particularly surprising. Hence, changing the INP concentration only has a much smaller effect on the cloud properties at the station. Over the ocean on the other hand, temperatures are still high enough that liquid water may persist under lower INP concentrations. The behaviour now resembles the summer period at PEA, indicating that it is mostly influenced by temperature, rather than location. This also means that, in winter, the change in INP concentration has more significant impacts on the cloud phase for the Southern Ocean, as with sufficient INPs, almost all liquid water will freeze. The response of liquid cloud water
mass to a change in INP concentration is about the same in both the summer and the winter period over the southern ocean, but the percentual change and the change in liquid mass fraction of hydrometeors is much larger during the winter. Thus, over the ocean, INP concentrations have a much more significant impact during the winter than during the summer, which is in contrast to the behaviour over land at PEA. This can also be seen in the CRE: The largest influence of INP concentration on total CRE is seen during the winter period, over the ocean, and while there is no influence of INP concentrations on total CRE during the
summer at PEA, such an influence can be seen on the individual components (LW and SW).

Finally, it should be noted that our model, while representing a wide range of INP concentrations, is limited to the parametrisations it uses. There is evidence that Secondary Ice Processes not represented in the model are temporally and spatially variable (Georgakaki et al., 2022), and that the ratio of Ice Crystal Number Concentration to the concentration of active INP is temperature-dependent (Järvinen et al., 2022), so increasing the INP concentration by a constant factor to account for missing
Secondary Ice Processes inevitably leads to errors. The distribution of activation temperatures, as prescribed in Formula 2, might also be a source of inaccuracy, as we have only tested one distribution and used a scaling factor for different INP concentrations. Other distributions exist, such as the "Marcus fit" presented in Vignon et al. (2021) for example, which does have any additional INPs activating in the lower temperature range below about $-30\,^\circ$C, while having a steeper increase in activated INPs between $-15\,^\circ$C and $-30\,^\circ$C. Another possible source for lower liquid water amounts could be the chosen low CCN
concentration, as the resulting lower droplet number concentration should reduce the amount of INPs required for nucleation. All in all, it is conceivable that the lack of representation of stratocumulus clouds with supercooled liquid water, which were



observed at the station, and the possible overestimation of INPs at lower temperatures lead to an overall underrepresentation of liquid water in the model. This would likely mean that the actual effects caused by changing INP concentrations are stronger than presented here, as the liquid water amounts in the lowest INP concentration would be likely be enhanced the most by the
inclusion of additional clouds.

## 6   Conclusions

Our results highlight the importance of ice nucleating particles (INPs) for the cloud phase in Antarctica. While the simulated clouds are not perfectly matching the observations at the station in terms of cloud phase, limiting the amount of available INPs does result in an increase of liquid water in clouds and is more closely in correspondance to the ceilometer, MRR, and INP
observations at the station. This effect is shown to be particularly relevant during the austral summer for continental Antarctica, whereas during austral winter, the colder temperatures facilitate homogeneous freezing and INP concentrations therefore become less important. Over the Southern Ocean, the opposite is the case: During austral summer, temperatures are high enough to allow liquid water to persist in the clouds at any INP concentration, whereas during the winter, a higher INP concentration leads to the complete freezing of clouds. The change in cloud phase also has radiative effects, but in the given model setup, an
improved representation of INPs would not alleviate biases in the near surface radiation. Further research is needed to improve the simulation, in particular the cloud phase, and with respect to microphysical processes that are not yet (well) represented in the model, such as secondary ice processes beyond Hallet-Mossop. The current version of the INP simulation module is computationally expensive, due to the 16 added variables, and a simplified and more optimised parametrisation might be sufficient. An increased vertical resolution might then help in alleviating the remaining model errors in the representation of clouds.
When restricting the CRE statistics to the thicker clouds, the radiative effects of the cloud were stronger in the cases where the concentration of INPs had relevant impacts (i.e. during summer over the continent, during winter over the ice sheet).

## Appendix

*Code and data availability.*   The ceilometer observations are available at https://doi.org/10.48804/07SS6R. The MRR observations are available at https://doi.org/10.48804/MDDKU0. The model output data required for recreating the figures presented in this article are available at
https://doi.org/10.48804/XGJVIZ. The model source code is available upon request.

*Author contributions.*   FS created the simulations, lead the analysis and the writing of the paper, and maintains the ceilometer and MRR observational datasets. NS implemented the used version of the model, and previously maintained the ceilometer and MRR datasets. AP helped with the implementation of the model and created the aerosol module. PvO collected the aerosol samples and helped with the



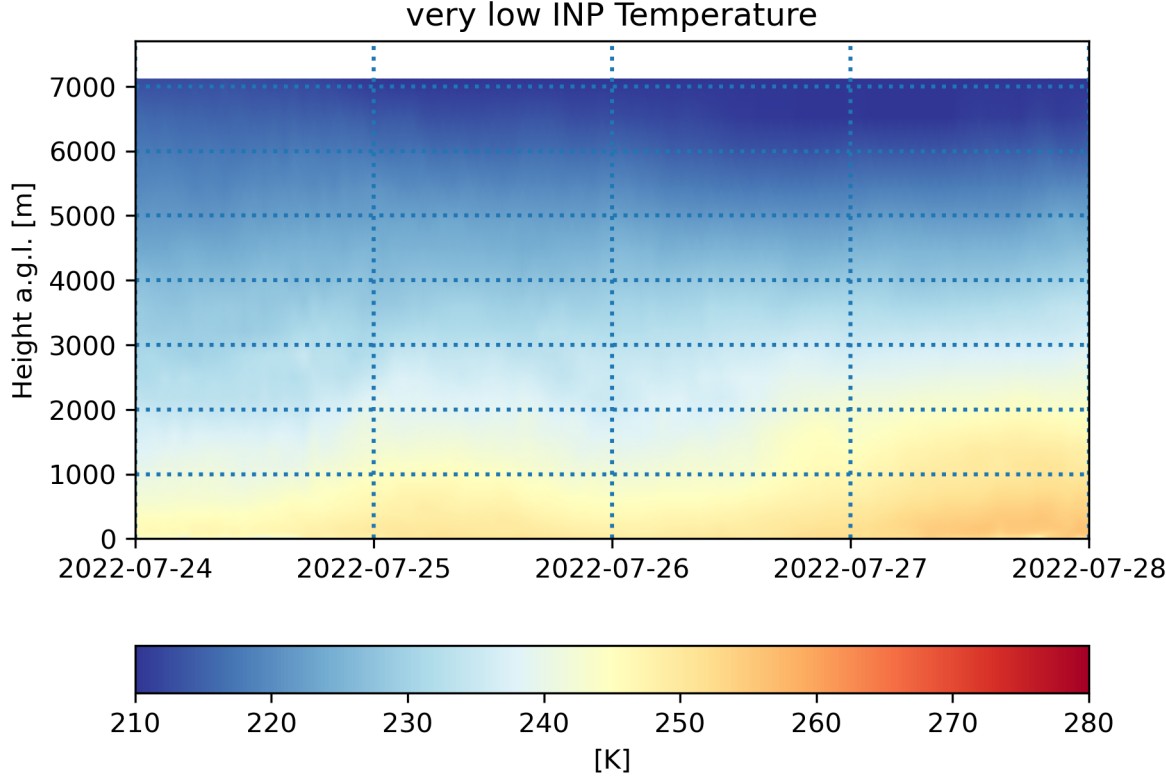

**Figure A1.** Temperature modelled at PEA (Princess Elisabeth Station, Antarctica) in the time period 24 July 2022 to 28 July 2022 for the VL (very low INP concentration, $1 \times 10^{-5}\,\mathrm{L^{-1}}$ at $-20\,°\mathrm{C}$) concentration.

maintenance of the ceilometer and MRR. HW analysed the aerosol samples. AM and NvL acquired the funding. NS, AP, HW, AM, KvW, 
and NvL helped with the analysis and the interpretation of the results. All authors helped in revising the paper and agreed to the final version.

*Competing interests.* The authors declare that they have no conflicts of interest.

*Acknowledgements.* We would like to thank Alexandra Gossart and Irina Gorodetskaya, as well as the International Polar Foundation (http://www.polarfoundation.org/) for their contributions in setting up and maintaining the MRR and Ceilometer. Furthermore, we would like to thank Maximilian Maahn for the creation of IMProToo, which was used for postprocessing the MRR data (Maahn and Kollias, 2012). 
We thank the COSMO-CLM community for creating and maintaining the COSMO-CLM regional climate model. The resources and services used in this work were provided by the VSC (Flemish Supercomputer Center), funded by the Research Foundation - Flanders (FWO) and the Flemish Government. This work was financed by the Belgian Federal Public Planning Service Science Policy, Brain-be 2.0 programme, project number B2/191/P1/CLIMB entitled "How do aerosol-CLoud Interactions influence the surface Mass Balance in East Antarctica?".



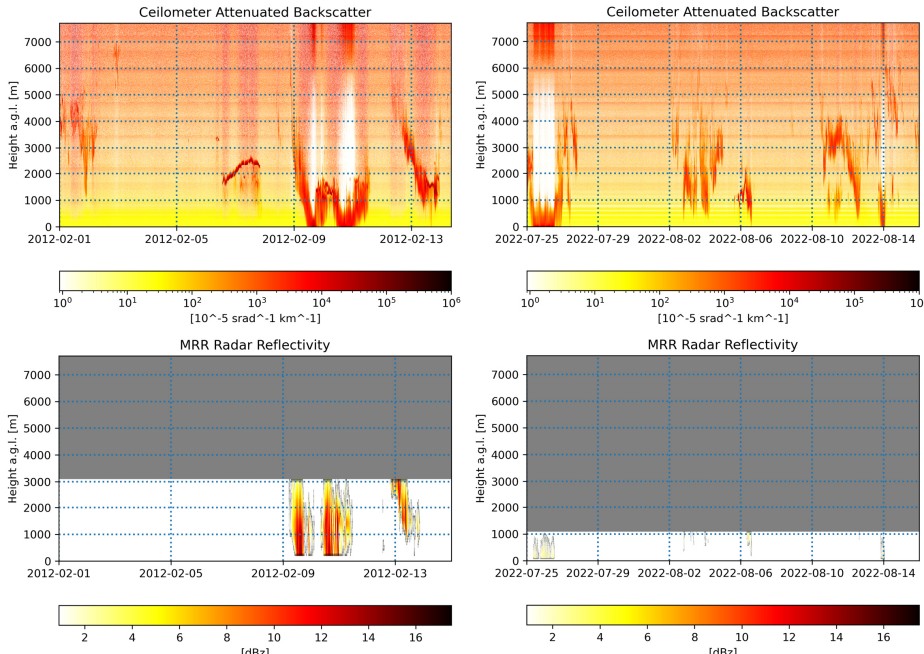

**Figure A2.** Ceilometer (top) and MRR (bottom) measurements for the time periods 1 February 2012 to 15 February 2012 (left) and 25 July 2022 to 16 August 2022 (right). The MRR was set to only measure up to a height of 1 km at an increased resolution during the winter period (right).

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
