# Peer review of "Ice-nucleating particle concentration impacts cloud properties over Dronning Maud Land, East Antarctica, in COSMO-CLM2"

_EGUsphere, 2024_

## Author Comment (AC1)

**Response to the reviewers**

21 August 2024

We thank the reviewers for their assessment and ideas to improve our work. In the following, we address their suggestions point by point, which has allowed us to remove some weaknesses our manuscript still had.

The authors present an interesting and relevant study on the impact of modeled INP concentrations on cloud properties in the Antarctic. They assess both the impact on cloud phase and cloud radiative effect. This is a valuable contribution to model development and improvement for polar regions with typically low INP concentrations. Overall, the study holds a very good scientific standard, but the presentation (writing and figures) has shortcomings and should be improved.

**Specific Comments**

**Comment 1** — Page 1-2, line 19-22: This part sounds very specific and slightly simplifying to come that early in the introduction, as there exist other cloud structures than the two-layered one as well. Please consider writing something about the occurrence and presence of mixed-phase clouds in Antarctica in general in this prominent position. This very recent paper by Dietel et al. and references therein might have some inspiration: https://acp.copernicus.org/articles/24/7359/2024/

**Reply** — We acknowledge that additional information should be given on the general cloud type distribution over the Southern Ocean and Antarctica. We therefore modified the text and included the recommended reference:

- Microphysical properties of mixed-phase clouds are heavily influenced by ice nucleating particles [INPs; Kanji et al., 2017], which grow efficiently via depositional growth [Korolev, 2007, Morrison et al., 2012]. In Antarctica, where INPs are very sparse compared to mid-latitude regions, the number of cloud droplets freezing at temperatures above about $-37\,$C, the temperature at which water freezes homogeneously [Murray et al., 2010], is therefore limited. This leads to the presence of liquid cloud droplets in a supercooled state alongside ice crystals. Over the Southern Ocean, liquid-containing clouds including those mixed-phase clouds (MPC) make up the majority of clouds below a height of 4-7 km [Dietel et al., 2024]. Over Antarctica, liquid-containing clouds were observed during 20 % of overcast periods [Gorodetskaya et al., 2015]. MPCs can often be identified by their characteristic two-layer structure, with a liquid layer above

an ice layer, which is generated by faster growth and larger sedimentation velocities of the relatively few ice crystals near cloud top [Bromwich et al., 2012].

**Comment 2** — Page 2, line 29-32: I suggest rephrasing these two sentences. You may add that you compare a liquid water containing cloud and an ice-only cloud having the same total water content (at least I assume this is what you mean), and possibly a short statement about why that is. The second sentence (As a result [. . . ] ice-water conversion.) is currently very vague and might be formulated more specific or eventually omitted.

**Comment 3** — Page 2, line 32: Depending on the rephrasing of the previous sentences, make sure that it is clear what "those effects" refers to or reformulate. Also, there might be other potential changes beyond cloud phase that affect uncertainty in global climate models. If "those effects" also should include potential changes in cloud cover, total cloud water content etc. please state that."

**Reply** — We edited the paragraph and added an additional reference. The sentence referred to in Comment 3 was removed as the cited papers were referring to different cloud feedbacks that are not relevant for this paragraph.

- At equal total water content, clouds containing supercooled liquid water have a larger CRE as liquid droplets reflect more sunlight, decreasing $SW_{all-sky}^{net}$, but they also are more opaque to longwave radiation, increasing $LW_{all-sky}^{net}$ [Matus and L'Ecuyer, 2017]. This implies that changes in cloud phase caused by a change in INP concentration can have a significant secondary effect in the change in radiative forcing, and therefore, the surface energy balance. This in turn can have other effects, for example, the cloud phase has a significant impact on the rate of surface melt [Gilbert et al., 2020].

**Comment 4** — Page 2, line 36/37: Please consider adding against which kind of observations the bias is expressed (satellite observations, I assume).

**Reply** — We added the requested information to the text:

- In most weather and climate models, there is a positive bias in the net SW radiation ($SW_{all-sky}^{net}$) over the Southern Ocean **compared to observations derived from the CALIPSO missions (CloudSat/Cloud–Aerosol Lidar and Infrared Pathfinder Satellite Observations)** [Kay et al., 2016].

**Comment 5** — Page 3, line 55: Please add some information about the location of Dome C, either in coordinates or in words.

**Reply** — We added the coordinate to the text:

- There have been attempts to reduce the radiation biases through correcting the liquid water content: Supercooled liquid water clouds that were observed using the station's instruments at Dome C (75.10°S, 123.35°E)

were modelled in two case studies using the regional climate model (RCM) ARPEGE-SH [Action de Recherche Petite Echelle Grande Echelle – Southern Hemisphere; Ricaud et al., 2020].

**Comment 6** — Page 3, line 59-61: Please add a reference (or move the reference Kretzschmar et al. 2020 to this sentence, if it is the same).

**Reply** — This reference indeed should go with the first sentence, as is now the case in the revised text.

- In ICON (icosahedral nonhydrostatic weather and climate model), a bias in SW radiation balance was found to be caused by an underestimation of the cloud layer's thickness, liquid water content, and hydrometeor number concentration [Kretzschmar et al., 2020]. Changing the cloud condensation nuclei (CCN) activation scheme reduced the bias in ICON, but did not fully resolve it.

**Comment 7** — Page 4, line 92-95: This sentence is very long and a bit complicated. I suggest splitting and rephrasing.

**Comment 8** — I also suggest changes to this sentence: What is meant by "given their focus on these regions"? Climate and weather models do not automatically have a focus on polar regions.

**Reply** — We rephrased the section to make it more readable. "These regions" should refer to the mid-latitude regions, which is what we expect the models are optimized for. We adapted that part to make our point more clear.

- In this paper, we test if there is significant variation between INP concentrations that are relevant for Antarctica today, using unique INP measurements from PEA. Such a sensitivity to Antarctic range INPs would indicate a need for a detailed simulation of INP in climate and weather models. Furthermore, we test if there is a significant impact to be expected when using INP concentrations measured in mid-latitude regions, which is relevant for evaluating climate and weather models. These models are most frequently applied to mid-latitude regions, and therefore parametrisations for ice nucleation are likely optimised to match the effects of higher INP concentrations found there. It might also become relevant should INP concentrations in Antarctica change in the future: Twohy et al. (2021) suggest that a decrease in Antarctic Sea Ice and an increase of water temperatures in the southern ocean could result in an increase of INP concentrations in Antarctica.

**Comment 9** — Page 4, line 98: Remove "can", just write "we assess".

**Reply** — We adapted the text according to the suggestion.

**Comment 10** — Page 4-5, line 112-122: Ice nucleation is often described divided into different mechanisms (contact freezing, deposition freezing, immersion freezing). Could you elaborate on which mode your measurements belong to? As the module used in your study is different from the scheme used by

Vignon et al. in WRF e.g., this information is not crucial to your model experiments, but it would be interesting to know for the reader to be able to compare to other studies.

**Reply** — While a separate paper about the measurements is still in preparation, we added the requested information to the text, alongside some other details on the measurement technique in response to the remarks of Reviewer 2:

- In addition to the weather and cloud observations, ground-based INP measurements were taken in the 2020/21 and 2021/22 austral summers. These INP measurements were taken using 47 mm polytetrafluoroethylene filters with a pore size of 800 nm (Whatman Nuclepore No. 10417312), which were set up in a shelter around 500 m north of PEA station. The 47 mm filters were placed inside a hard plastic filter holder. This had a metal cap (inversed funnel type) with an inlet opening of 0.25" diameter. On it, a 15 cm piece of black conductive silicon tubing with a 0.25" outer diameter and 0.19" inner diameter was fitted. The filter holder was situated outside, located 50 cm above the shelter's roof. The end of the 15 cm conductive tubing pointed downward and was oriented perpendicular to the main wind direction (NE). The sampling losses within the 15 cm tube are negligible at the given flow rate and with particles larger than 1 µm being very rare in the PEA area [Herenz et al., 2019]. Sample duration was around 10 days per filter, and each season, blank samples were taken. The subsequent measurements were done in the same way as in Sze et al. [2022], using the two well-established off-line techniques LINA (Leipzig ice nucleation array) and INDA [ice nucleation droplet array; Lacher et al., 2024]. The INP profiles of the blank samples were substracted from the measurement results, although the difference compared to the results derived from the samples directly was very small. Our observations at PEA are compared here with observations taken from literature in order to identify suitable INP concentrations to use for the sensitivity experiments performed with COSMO-CLM².

**Comment 11** — Page 6, line 165-166: I suggest changing the sentence to the following (please check that this equally correct):

This module adds 16 different INP concentration variables corresponding to different activation temperatures.

**Reply** — This is a clearer formulation, we therefore implemented the suggested change to the text.

**Comment 12** — Page 7, line 188-189: "Integration" does not sound like the right word to use here. Do you mean simulation, implementation, version or something else?

**Reply** — Indeed, we replaced "integration" with "simulation" throughout the manuscript.

**Comment 13** — Page 7, line 188-195: This whole part is important, but not really suitable in the section Model description. It would fit better either in

the end of the Introduction or at the beginning of the Results section. Please consider adapting and moving it.

**Reply** — We agree thath this would be a good start of the results section with some changes, as also suggested in Comment 16. Therefore, the section was removed here and added to the results part (see Comment 16).

**Comment 14** — Page 8, line 209: You might also mention Schäfer et al. 2024 https://acp.copernicus.org/articles/24/7179/2024/ during the discussion of SIP, as their study (although for the Arctic) included observationally constrained INP concentrations addition to added SIP (again relevant for Discussion part as well).

**Reply** — Thank you for recommending this very interesting read! We added the reference to the section:

- This medium concentration also serves as an augmentation for the SIP modes not captured by our setup (except rime splintering which is included), as it was found that these modes can increase ice crystal number concentrations (ICNC) by a factor of 10 [Sotiropoulou et al., 2020] in Antarctica. It is noted that this assumption is not very accurate: SIPs are only active very locally with great spatial variance [Georgakaki et al., 2022], and simulating additional SIP modes are necessary for a realistic simulation of the vertical structure of MPCs [Schäfer et al., 2024]. Due to the scale of the model however, we expect the overall error on caused by this to be small.

**Comment 15** — Page 8, line 216-220: For which time periods and cloud types/altitudes were the tests performed? Can you tell whether they covered the environmental conditions present throughout the total study?

**Reply** — We agree that the description of these tests can be extended. Please find the adapted text underneath:

- However, we performed initial tests, varying the prescribed CCN concentrations from $10\,\mathrm{cm}^{-3}$ to $1300\,\mathrm{cm}^{-3}$, corresponding to the measured range at PEA [Herenz et al., 2019]. These tests were performed for the time period from 03 January 2016 to 12 January, which featured similar weather conditions as the austral summer period presented here. The results showed, agreeing with previous findings [Solomon et al., 2018], a negligible impact of CCN concentrations on cloud phase, which is why the impact of CCN concentrations was not investigated further. Thus, in all of our simulations, we used the low-end CCN concentration of $10\,\mathrm{cm}^{-3}$.

**Comment 16** — Page 9, line 242: The beginning of the Results section should be improved. What do you mean by "In the simulation", possibly "In our simulations/experiments"? As indicated earlier, some of the discussion of the end of 3.1 might fit here and it would be useful to introduce the structure of the Results section to the reader.

**Reply** — Indeed, the introduction to the results section can be improved. Fortunately, the paragraph mentioned in Comment 13 is a good fit with some adaptations. We therefore moved it here:

- By comparing the results of a simulation where we prescribe an INP concentration on the low end of the observed range in Antarctica to those of a simulation where we prescribe a concentration on the high end, we tested the model sensitivity of clouds and radiation to different INP concentration. Furthermore, the results achieved with Antarctic INP concentrations are compared to those with mid-latitude concentrations, which are unrealistically high for the region. Finally, all of this output is compared to cloud observations taken at PEA to verify that cloud properties are well-represented and to get insight in what INP concentrations give the most realistic results. Overall, the results show a strong connection between INP concentration and cloud liquid water contents.

**Comment 17** — Page 10, line 246: What do you mean by frontal cloud? If you mean a certain time period or altitude region, please indicate that. If you mean the whole (shown) cloud, I suggest removing "frontal".

**Reply** — We used the "frontal" to identify the cloud within the entire simulation period (see line 229), but as only this cloud is shown in Figure 2, the word can indeed be removed for more clarity.

- This behaviour can be seen in Fig. 2, where a significant portion of hydrometeors in the cloud remain in a liquid state when using the VL setting, as opposed to the VH setting.

**Comment 18** — Page 10, line 260: The terms VL concentration and VL setting are used aside throughout the manuscript, sometimes also VL INP concentration. I recommend choosing one nomenclature and sticking to it.

**Reply** — This is indeed confusing. We decided to go with "setting" in order to have a better distinction between the concrete setting used and actual concentrations, and changed all occurrences accordingly.

**Comment 19** — Page 11, line 267: The terms TQC and TQI are nowhere explained in the text apart from figure captions. Please add or remove the term in brackets.

**Reply** — We added an explanation to the terms:

- In other words, the liquid mass fraction of hydrometeors (i.e., $\frac{TQC}{TQI+TQC}$, with TQC being the vertically integrated liquid water content, TQI being the vertically integrated ice water content) at PEA in the summer period increased to 12.6 % in VL, from 1.3 % in VH.

**Comment 20** — Page 14, line 298-301: Change beginning of the sentence to: When applying the constraint to only count timesteps with at least [. . .]. I also suggest making the last part starting with "highlighting the stronger" a new sentence

**Reply** — We adapted the suggested change:

- When applying the constraint to only count timesteps with at least 0.15 $gm^{-2}$ of liquid water available, it can be seen that the radiative effects of

these clouds are much stronger, placing the medians between -45 Wm$^{-2}$ and -50 Wm$^{-2}$ (VL: -46.3 Wm$^{-2}$, L: -45.5 Wm$^{-2}$, M: -47.9 Wm$^{-2}$, H: -45.9 Wm$^{-2}$, VH: -48.0 Wm$^{-2}$). This also highlights how liquid cloud water causes stronger cloud radiative effects.

**Comment 21** — Page 14, line 304: Switch order of VH and M in text to match order in figures.
**Reply** — We adapted the text according to the suggestion:

- During winter, the median CRE is increased in the VL setting compared to M and VH (Fig. 6g: VL: 81.6 Wm$^{-2}$, M: 79.8 Wm$^{-2}$, VH: 79.4 Wm$^{-2}$) over the ocean, using the more relaxed condition of at least 0.05 gm$^{-2}$ of water and ice.

**Comment 22** — Page 15, line 308-310: This sentence should be improved, potentially avoiding the repetition of close to 0 Wm-2 and instead repeating what the increase refers to (from VL to VH).
**Reply** — We changed the sentence for more clarity:

- As Fig. 6g shows, the increase in CRE in the VL setting is more pronounced when comparing the the timesteps with a smaller CRE. In the VH setting, there are a lot of samples clustered at or slightly above a CRE of 0 Wm$^{-2}$, while that cluster spreads out to 20-40 Wm$^{-2}$ for the M and VL settings.

**Comment 23** — Page 17, line 359-360: I am not sure if I understand what you want to convey with this sentence. The prescribed CCN concentration may also influence vertical cloud structure through different size distributions and sedimentation velocities. But mainly, I am again wondering about the conditions of the tests you performed and wrote about earlier. If they covered all possible conditions and you didn't see an influence on liquid water content, the CCN concentrations might really not be of importance, but if you still consider the prescribed (low) CCN concentration a relevant source of error/uncertainty, some more discussion/references might be good. Please clarify.
**Reply** — As the influence of CCN in our model simulations is close to zero, so we decided to remove the sentence altogether.
**Comment 24** — Page 18, line 363: Could you use a stronger formulation than "could likely mean"?
**Reply** — We changed the text and now use "implies" instead of "would likely mean".
**Comment 25** — Page 18, line 377: Please replace "Hallet-Mossop" by "rime splintering". ("Hallet-Mossop" has never been introduced as a term, although you correctly cite them in line 174.)
**Reply** — We adapted the text according to the suggestion.
**Comment 26** — Page 18, line 379: I suggest replacing "alleviating" by "reducing".
**Reply** — We adapted the text according to the suggestion.

**Comment 27** — Page 18, line 380: I assume that by thicker clouds you mean the clouds where the water content exceeds your threshold. Consider a better formulation so that it can't be confused with geometrically thicker.

**Reply** — This is indeed what we meant. We adapted the text to reflect this circumstance better:

- When restricting the CRE statistics to optically denser clouds, the radiative effects of the cloud were stronger in the cases where the concentration of INPs had relevant impacts (i.e. during summer over the continent, during winter over the ice sheet).

**Comment 28** — Figures 2 and 3: Please add panel labels (a), (b) etc. to the individual panels and use them when referring to the figures in the text. For the ceilometer colorbar: Please write the exponents as exponents and not ^1. You might consider removing the long headings for the six individual plots and instead add the variable (Attenuated Backscatter, Radar Reflectivity, Cloud Ice Content, Cloud Liquid Water Content) to the x-axis label and the instrument/simulation name (Ceilometer, MRR, VL, VH) in the upper left corner of each plot."

**Reply** — We adapted the figures to include the suggested changes:

[Figure]

Figure 2: Cloud liquid water (bottom row) and ice (top row) for the VL (very low INP concentration, $1 \times 10^{-5}\,\mathrm{L}^{-1}$; middle column) and VH (very high INP concentration, $2\,\mathrm{L}^{-1}$; right column) settings, compared to the ceilometer (top left) and MRR (bottom left) measurements, in the time period 8 August 2012 to 12 August 2012.

**Comment 29** — Figure A1: Please remove "Temperature" from the heading and put it into the colorbar label instead.

**Reply** — We adapted the figure to include the suggested change:

**Comment 30** — Figure A2: Same comment as to Figure 2 and 3 applies, but is less important in the appendix.

**Reply** — We still adapted the figure to include the suggested changes:

**Technical Corrections**

**Comment 31** — Title/page 4, line 101: The term "Dronning Maud Land" appears only in the title and then never again. Instead, the term "Queen Maud Land" appears in the first sentence of the Observations part. Please make that consistent and use the same term both places.

[Figure]

Figure 3: Cloud liquid water (bottom row) and ice (top row) for the VL (very low INP concentration, $1 \times 10^{-5}\,\text{L}^{-1}$ at $-20\,\text{C}$; middle column) and VH (very high INP concentration, $2\,\text{L}^{-1}$ at $-20\,\text{C}$; right column) settings, compared to the ceilometer (top left) and MRR (bottom left) measurements, in the time period 24 July 2022 to 28 July 2022. The MRR was set to only measure up to a height of 1 km at an increased resolution for this time period.

[Figure]

Figure A1: Temperature modelled at PEA (Princess Elisabeth Station, Antarctica) in the time period 24 July 2022 to 28 July 2022 for the VL (very low INP concentration, $1 \times 10^{-5}\,\text{L}^{-1}$ at $-20\,\text{C}$) setting.

[Figure]

Figure A2: Ceilometer (top) and MRR (bottom) measurements for the time periods 1 February 2012 to 15 February 2012 (left) and 25 July 2022 to 16 August 2022 (right). The MRR was set to only measure up to a height of 1 km at an increased resolution during the winter period (right).

**Reply** — We changed "Queen Maud Land" to "Dronning Maud Land".

**Comment 32** — Page 4, line 97: Please correct spelling: Antarctic sea ice (no capital letters in sea ice).

**Reply** — We adapted the text according to the suggestion.

**Comment 33** — Page 4, line 98: Please correct spelling: Southern Ocean (with capital letters at the beginning).

**Reply** — We adapted the text according to the suggestion.

**Comment 34** — Page 5, line 147: Correct spelling: ground-based lidar.

**Reply** — We adapted the text according to the suggestion.

**Comment 35** — Page 6, line 162: Please correct spelling: Antarctic ice sheet (no capital letters in ice sheet).

**Reply** — We adapted the text according to the suggestion.

**Comment 36** — Page 7, line 174: Please correct spelling: rime splintering (no capital letters).

**Reply** — We adapted the text according to the suggestion.

**Comment 37** — Page 7, eq. 2: There is one redundant closing bracket at the end of the equation. You might also consider to write the exponent in superscript instead of in brackets.

**Reply** — We removed the bracket, which was indeed redundant. We would like to keep the exponent written out the way it is in order to be consistent with the original paper.

**Comment 38** — Page 8, line 210: What is the plural "SIPs" meaning here – secondary ice productions? I suggest writing SIP modes instead.

**Reply** — We adapted the text according to the suggestion.

**Comment 39** — Page 8, line 214: Citation Bigg and Hopwood 1963 should be without brackets.

**Reply** — We adapted the text according to the suggestion.

**Comment 40** — Figure 1: Consider moving the labels (a), (b), (c) in the caption text in front of the dates instead of behind.

**Reply** — We adapted the text according to the suggestion.

**Comment 41** — Figures 4, 5, 6: Units on axis labels should not be italic.

**Reply** — We adapted the figures according to the suggestion.

**Comment 42** — Page 14, line 298: Correct spelling: with.

**Reply** — We adapted the text according to the suggestion.

**Comment 43** — Page 16, line 311: Please correct spelling: optically.

**Reply** — We adapted the text according to the suggestion.

**Comment 44** — Page 17, line 327: Citation Ricaud et al. 2024 should be without brackets.

**Reply** — We adapted the text according to the suggestion.

**Comment 45** — Page 17, line 353: Correct spelling: ice crystal number concentration (no capital letters).

**Reply** — We adapted the text according to the suggestion.

**Comment 46** — Page 17, line 352, 355: Please correct spelling: secondary ice processes (no capital letters). Or use the abbreviation SIP that you introduced for secondary ice production earlier. As you probably notice, I commented

on the spelling regarding small or capital letters quite a few places and I noticed that this was not consistent for many terms throughout the manuscript. I recommend going through the whole manuscript carefully checking which terms to use capital letters for and which not, as I might have overseen some places.

**Reply** — We adapted the text according to the suggestion and use "SIP modes" now. Upon checking the text for other capitalisation issues, we corrected a small number of additional spelling and grammar issues. We also found that replacing "ocean" with "Southern Ocean" was appropriate in some places.

**Comment 47** — Page 17, line 357: MARCUS is an acronym for a campaign. It should be written with capital letters only and the long name should also be given.

**Reply** — We adapted the text according to the suggestion, the full campaign name is now added in brackets.

**Comment 48** — References: Many references are missing clickable doi or url links which makes it harder to quickly click and check something for the reader/reviewer. This is just meant as a tip that it is nice to provide the links from the beginning and not first when the copyediting/typesetting office demands it.

**Reply** — We have checked all references and added urls where required. For one paper [Lacher et al., 2024], we replaced the preprint with the final version.

**References**

Zamin A. Kanji, Luis A. Ladino, Heike Wex, Yvonne Boose, Monika Burkert-Kohn, Daniel J. Cziczo, and Martina Krämer. Overview of ice nucleating particles. *Meteorological Monographs*, 58:1.1 – 1.33, 2017. doi: 10.1175/AMSMONOGRAPHS-D-16-0006.1. URL https://journals.ametsoc.org/view/journals/amsm/58/1/amsmonographs-d-16-0006.1.xml.

Alexei Korolev. Limitations of the wegener–bergeron–findeisen mechanism in the evolution of mixed-phase clouds. *Journal of the Atmospheric Sciences*, 64(9):3372 – 3375, 2007. doi: 10.1175/JAS4035.1. URL https://journals.ametsoc.org/view/journals/atsc/64/9/jas4035.1.xml.

Hugh Morrison, Gijs De Boer, Graham Feingold, Jerry Harrington, Matthew D Shupe, and Kara Sulia. Resilience of persistent arctic mixed-phase clouds. *Nature Geoscience*, 5(1):11–17, 2012. doi: 10.1038/ngeo1332. URL https://doi.org/10.1038/ngeo1332.

B. J. Murray, S. L. Broadley, T. W. Wilson, S. J. Bull, R. H. Wills, H. K. Christenson, and E. J. Murray. Kinetics of the homogeneous freezing of water. *Phys. Chem. Chem. Phys.*, 12:10380–10387, 2010. doi: 10.1039/C003297B. URL http://dx.doi.org/10.1039/C003297B.

B. Dietel, O. Sourdeval, and C. Hoose. Characterisation of low-base and mid-base clouds and their thermodynamic phase over the southern ocean and arctic marine regions. *Atmospheric Chemistry and*

*Physics*, 24(12):7359–7383, 2024. doi: 10.5194/acp-24-7359-2024. URL https://acp.copernicus.org/articles/24/7359/2024/.

Irina V Gorodetskaya, Stefan Kneifel, Maximilian Maahn, Kristof Van Tricht, Wim Thiery, JH Schween, Alexander Mangold, Susanne Crewell, and NPM Van Lipzig. Cloud and precipitation properties from ground-based remote-sensing instruments in east antarctica. *The Cryosphere*, 9(1):285–304, 2015. doi: 10.5194/tc-9-285-2015. URL https://tc.copernicus.org/articles/9/285/2015/.

David H Bromwich, Julien P Nicolas, Keith M Hines, Jennifer E Kay, Erica L Key, Matthew A Lazzara, Dan Lubin, Greg M McFarquhar, Irina V Gorodetskaya, Daniel P Grosvenor, et al. Tropospheric clouds in antarctica. *Reviews of Geophysics*, 50(1), 2012. doi: 10.1029/2011RG000363. URL https://doi.org/10.1029/2011RG000363.

Alexander V. Matus and Tristan S. L'Ecuyer. The role of cloud phase in earth's radiation budget. *Journal of Geophysical Research: Atmospheres*, 122(5):2559–2578, 2017. doi: 10.1002/2016JD025951. URL https://agupubs.onlinelibrary.wiley.com/doi/abs/10.1002/2016JD025951.

E. Gilbert, A. Orr, J. C. King, I. A. Renfrew, T. Lachlan-Cope, P. F. Field, and I. A. Boutle. Summertime cloud phase strongly influences surface melting on the larsen c ice shelf, antarctica. *Quarterly Journal of the Royal Meteorological Society*, 146(729):1575–1589, 2020. doi: 10.1002/qj.3753. URL https://rmets.onlinelibrary.wiley.com/doi/abs/10.1002/qj.3753.

Jennifer E. Kay, Casey Wall, Vineel Yettella, Brian Medeiros, Cecile Hannay, Peter Caldwell, and Cecilia Bitz. Global climate impacts of fixing the southern ocean shortwave radiation bias in the community earth system model (cesm). *Journal of Climate*, 29 (12):4617 – 4636, 2016. doi: 10.1175/JCLI-D-15-0358.1. URL https://journals.ametsoc.org/view/journals/clim/29/12/jcli-d-15-0358.1.xml.

P. Ricaud, M. Del Guasta, E. Bazile, N. Azouz, A. Lupi, P. Durand, J.-L. Attié, D. Veron, V. Guidard, and P. Grigioni. Supercooled liquid water cloud observed, analysed, and modelled at the top of the planetary boundary layer above dome c, antarctica. *Atmospheric Chemistry and Physics*, 20(7):4167–4191, 2020. doi: 10.5194/acp-20-4167-2020. URL https://acp.copernicus.org/articles/20/4167/2020/.

J. Kretzschmar, J. Stapf, D. Klocke, M. Wendisch, and J. Quaas. Employing airborne radiation and cloud microphysics observations to improve cloud representation in icon at kilometer-scale resolution in the arctic. *Atmospheric Chemistry and Physics*, 20 (21):13145–13165, 2020. doi: 10.5194/acp-20-13145-2020. URL https://acp.copernicus.org/articles/20/13145/2020/.

Paul Herenz, Heike Wex, Alexander Mangold, Quentin Laffineur, Irina V Gorodetskaya, Zoë L Fleming, Marios Panagi, and Frank Stratmann. Ccn measurements at the princess elisabeth antarctica research station during three austral summers. *Atmospheric Chemistry and Physics*, 19(1):275–294, 2019.

Kevin Cheuk Hang Sze, Heike Wex, Markus Hartmann, Henrik Skov, Andreas Massling, Diego Villanueva, and Frank Stratmann. Ice nucleating particles in northern greenland: annual cycles, biological contribution and parameterizations. *Atmospheric Chemistry and Physics Discussions*, 2022:1–45, 2022. doi: 10.5194/acp-23-4741-2023. URL https://acp.copernicus.org/articles/23/4741/2023/.

L. Lacher, M. P. Adams, K. Barry, B. Bertozzi, H. Bingemer, C. Boffo, Y. Bras, N. Büttner, D. Castarede, D. J. Cziczo, P. J. DeMott, R. Fösig, M. Goodell, K. Höhler, T. C. J. Hill, C. Jentzsch, L. A. Ladino, E. J. T. Levin, S. Mertes, O. Möhler, K. A. Moore, B. J. Murray, J. Nadolny, T. Pfeuffer, D. Picard, C. Ramírez-Romero, M. Ribeiro, S. Richter, J. Schrod, K. Sellegri, F. Stratmann, B. E. Swanson, E. S. Thomson, H. Wex, M. J. Wolf, and E. Freney. The puy de dôme ice nucleation intercomparison campaign (picnic): comparison between online and offline methods in ambient air. *Atmospheric Chemistry and Physics*, 24(4):2651–2678, 2024. doi: 10.5194/acp-24-2651-2024. URL https://acp.copernicus.org/articles/24/2651/2024/.

G. Sotiropoulou, S. Sullivan, J. Savre, G. Lloyd, T. Lachlan-Cope, A. M. L. Ekman, and A. Nenes. The impact of secondary ice production on arctic stratocumulus. *Atmospheric Chemistry and Physics*, 20(3):1301–1316, 2020. doi: 10.5194/acp-20-1301-2020. URL https://acp.copernicus.org/articles/20/1301/2020/.

Paraskevi Georgakaki, Georgia Sotiropoulou, and Athanasios Nenes. Parameterizing secondary ice production in arctic mixed-phase clouds. In *EGU General Assembly Conference Abstracts*, pages EGU22–11263, 2022. doi: 10.5194/egusphere-egu22-11263. URL https://doi.org/10.5194/egusphere-egu22-11263.

B. Schäfer, R. O. David, P. Georgakaki, J. T. Pasquier, G. Sotiropoulou, and T. Storelvmo. Simulations of primary and secondary ice production during an arctic mixed-phase cloud case from the ny-ålesund aerosol cloud experiment (nascent) campaign. *Atmospheric Chemistry and Physics*, 24(12):7179–7202, 2024. doi: 10.5194/acp-24-7179-2024. URL https://acp.copernicus.org/articles/24/7179/2024/.

Amy Solomon, Gijs de Boer, Jessie M Creamean, Allison McComiskey, Matthew D Shupe, Maximilian Maahn, and Christopher Cox. The relative impact of cloud condensation nuclei and ice nucleating particle concentrations on phase partitioning in arctic mixed-phase stratocumulus clouds. *Atmospheric Chemistry and Physics*, 18

(23):17047–17059, 2018. doi: 10.5194/acp-18-17047-2018. URL
https://acp.copernicus.org/articles/18/17047/2018.

---

## Author Comment (AC2)

**Response to the reviewers**

August 2024

We thank the reviewers for their assessment and ideas to improve our work. In the following, we address their suggestions point by point, which has allowed us to remove some weaknesses our manuscript still had.

The authors present the results of a modeling study aimed at understanding the controls exerted by ice nucleating particles (INPs) on liquid water contents of Antarctic / Southern Ocean mixed phase clouds. Winter and summer numerical modeling simulations of cloud systems were set up, using prescribed CCN and INP concentrations, and varying the latter to investigate the impacts. The runs are guided by observational case studies. As the authors note, the simulations are generally qualitatively representative of the observations but "the model fails to accurately represent the long-lived, dense liquid and mixed-phase layers observed at the (continental) station". Indeed this is an ongoing issue in the simulation of polar mixed phase clouds that this work cannot address. Further, radiative biases are present in the model, but are noted to be on the same order as those reported in prior modeling studies.

The first main finding of this work is that contrasting air temperatures over land and ocean led to different seasonal impacts. Cold wintertime temperatures over the content supported homogeneous freezing and thus variations in INP had little impact, a perhaps expected finding. Over the ocean in wintertime, temperature were moderate enough such that liquid water contents responded strongly to INP concentrations, whereas in summertime, temperature were warm enough to limit ice formation regardless of INP concentrations. The second main finding is that the simulated changes in liquid water content have only a small radiative effect. This result seems to indicate that concern over changes in the radiation balance in the Antarctic induced by changes in cloud phase might be overstated.

An aspect of this work that represents an advance over prior treatments is that an INP budget is included, that has 16 different temperature activation bins, enabling a more realistic representation than fixed-INP concentrations. However, the slope of the cumulative distribution is fixed, using the DeMott et al. (2010) parameterization.

Overall, there is a nice description of the modeling system which is a useful addition to the literature, and the work takes a methodical approach to explore sensitivities. I have the following suggestions for changes to the paper. A number of these questions relate to the INP observations, although in the end they are used primarily to provide bounds on the variation input to the model.

If those measurements are not described elsewhere then I suggest to improve that discussion so that this work can serve to report them.

**Comments**

**Comment 1** — Abstract: when citing the INP concentrations (line 3) please indicate these are at -20 C. and that the analysis method used refers to immersion freezing INPs.

**Reply** — Those are some important information. We adapted the sentence:

- Ice-nucleating particles (INPs) have an important function in the freezing of clouds, but are rare in East Antarctica. At the Belgian Princess Elisabeth Station, immersion freezing INP concentrations between $6 \times 10^{-6}\,\mathrm{L^{-1}}$ and $5 \times 10^{-3}\,\mathrm{L^{-1}}$ have been observed with an activation temperature of -20°C.

**Comment 2** — line 48: the Kay et al. (2012) reference is over a decade old; have there been updates to CESM that have reduced the cited bias?

**Reply** — There has indeed been some progress in CESM2, as can be seen in [Gettelman et al., 2020]. We added a sentence in the text to reflect that update.

- The community earth system model (CESM) has a 30 Wm$^{-2}$ (warm) bias in $CRE_{SW}$ and a -10 Wm$^{-2}$ (cold) bias in $CRE_{LW}$ over the Southern Ocean [Kay et al., 2012] in version 1. This has since been reduced in CESM2 with the community atmosphere model (CAM) version 6 [Gettelman et al., 2020].

**Comment 3** — line 113: what is the pore size of the filters? not much detail is provided about the INP observations. Does the implementation begin at -15 C because the samples were below the limit of detection at warmer temperatures? could the DeMott (2010) parameterization be extended to "fill in" some reasonable values? why not use the slope of measured INP spectra, instead of the slope of this fit based on the DeMott global (not polar) fit?

**Comment 4** — line 114: how is the inlet oriented? was there no precipitation shield? what sampling inlet losses might be expected (the dimensions of the tube are not provided)?

**Comment 5** — line 118: how were blank corrections handled?

**Reply** — We have reworked and expanded the measurement section in order to include more details, also in response to Reviewer 1's comment. A paper dedicated to the measurements is also currently in preparation. As for the inlet, we added some information about orientation and design in the section. There was no further specific precipitation shield or inlet size cut-off. Precipitation at PE is only snow or drifting/blowing snow. With a specific inlet, the risk of clogging is high during elevated wind speed periods. And heating is no option at the ambient temperatures, there would be immediate re-freezing, producing an ice cover. The simple inlet tubing provided therefore the best option.

- In addition to the weather and cloud observations, ground-based INP measurements were taken in the 2020/21 and 2021/22 austral summers. These INP measurements were taken using 47 mm polytetrafluorethylene filters with a pore size of 800 nm (Whatman Nuclepore No. 10417312), which were set up in a shelter around 500 m north of PEA station. The 47 mm filters were placed inside a hard plastic filter holder. This had a metal cap (inversed funnel type) with an inlet opening of 0.25" diameter. On it, a 15 cm piece of black conductive silicon tubing with a 0.25" outer diameter and 0.19" inner diameter was fitted. The filter holder was situated outside, located 50 cm above the shelter's roof. The end of the 15 cm conductive tubing pointed downward and was oriented perpendicular to the main wind direction (NE). The sampling losses within the 15 cm tube are negligible at the given flow rate and with particles larger than 1 µm being very rare in the PEA area [Herenz et al., 2019]. Sample duration was around 10 days per filter, and each season, blank samples were taken. The subsequent measurements were done in the same way as in Sze et al. [2022], using the two well-established off-line techniques LINA (Leipzig ice nucleation array) and INDA [ice nucleation droplet array; Lacher et al., 2024]. The INP profiles of the blank samples were substracted from the measurement results, although the difference compared to the results derived from the samples directly was very small. Our observations at PEA are compared here with observations taken from literature in order to identify suitable INP concentrations to use for the sensitivity experiments performed with COSMO-CLM².

As for the used parametrisation, indeed, the measured INP versus temperature profile was flatter at lower temperatures compared to the DeMott parametrisation we used, however, the measurements were not yet fully evaluated at the start of the modelling. Adding another profile would be an interesting addition, and we are planning to do that for our next publications, however, our focus here lies on the differences between different overall INP concentrations and not so much on the spectra itself. It indeed gives us too many INPs at lower temperatures, which is a point we addressed in the Discussion, we edited it in order to clarify that a bit more. The main advantage of using the DeMott-parametrisation is that it makes our results more comparable to previous papers.

- The distribution of activation temperatures, as prescribed in Eq. (2), might also be a source of inaccuracy, as we have only tested one distribution based on the parametrisation by DeMott et al. [2010] and used a scaling factor for different INP concentrations. Other distributions often have a lower increase in the INP concentration at lower temperatures, such as the "MARCUS fit" (Measurement of Aerosols, Radiation and Clouds over the Southern Ocean) presented in Vignon et al. [2021], which does not have any additional INPs activating in the lower temperature range below about $-30$ C, while having a steeper increase in activated INPs between $-15$ C and $-30$ C.

**Comment 6** — Section 3.1: how is the INP budget handled during model spin-up time (is such spin up time considered?) In other words, as INP are removed from the domain (although also regenerated from evaporating precipitation and advected in), are the INP concentrations during the analyzed period markedly different from the initial condition?

It would be interesting to see a timeline / contour plot showing the budget of INPs in the simulations, perhaps selecting the -20C point in the spectrum for this, and including a discussion of any influence on the findings.

**Reply** — We added a figure in the appendix that shows the INP concentration over time for the summer period in two of our cases and added a paragraph in the Discussion to address this valid concern. We expect spin-up issues to be low, thanks to the frequent exchange of air masses, so we did not include a specific spin-up time, however, we still focused our analysis on the time periods towards the end of the simulation.

- The spin-up time is expected to be low, due to the frequent and fast exchange of air masses in relation to the domain size. As can be seen in Fig. A3, INP concentrations drop slightly initially, but stay close underneath their prescribed concentration. There is a significant drop at the end of the simulation period, but this drop is likely not related to spin-up, as after 2 months of simulation, all initial air masses should have been exchanged. The L and H settings are very similar in their INP timeline too, indicating that the deviations in concentrations are caused by synoptic-scale weather systems and not spin-up errors.

[Figure]

Figure A3: Average INP concentration at -20°C in the L (low INP concentration, $5 \times 10^{-3}\,\mathrm{L}^{-1}$) setting (a) and H (high INP concentration, $2 \times 10^{-1}\,\mathrm{L}^{-1}$) setting (b) for a 21x21 area around PEA at a height of 2250m.

**Comment 7** — line 174: is rime splintering ever active in these simulations? This mechanism relies on the presence of specific hydrometeor types and of certain size. Evidence for secondary ice production (SIP) in Southern Ocean clouds has been published, so excluding relevant SIP processes might be a shortcoming of this work. I was confused by the statement on line 208 that increasing INPs was equivalent to representing some secondary ice production modes (although the authors note in line 201 "this is not a very accurate assumption" and again qualify this approach in the Discussion (line 355)).

**Reply** — While we do not have immediate evidence that rime splintering was active in the clouds we observed, our parametrisation for rime splintering is

generally active at temperatures between 265 and 270 K and does not require a specific kind of ice particles to be present. These temperatures were not found in the winter (as shown in Figure A1), but fairly common across the summer period at lower levels at PEA as well as over the ocean, so rime splintering was likely active. It might be restricted by our highest temperature INP activation temperature being -15°C however, as this way, rime splintering relies on preexisting ice, either imported from the model boundaries or by the surrounding air briefly cooling down below -15°C. This should be addressed in further model development and we added a short section to the last paragraph of the discussion. We also recognize that not implementing all relevant SIP processes is a limitation of our work, but we believe it is justifiable as we are only looking at the cloud sensitivity to INPs. While these effects can be changed by SIPs, we expect that the average effect will be caused by enhancing the ICNC by a factor of 10 as stated, which can be approximated by increasing the INP concentration by that factor.

- On the higher temperature end, having the highest INP activation temperature at -15°C is a simplification as well. The concentration of INPs activating at such higher temperatures is extremely small and would likely have no measurable effect. However, not having any INPs means that rime splintering, which is active in the temperature range between -3°C and -8°C, has to rely on small amounts of ice already existing, as there is no primary ice nucleation active in that temperature range that could initiate secondary ice production. Possibly, this would increase the effects of INP concentration over the ocean in summer, which we found to be very low, as such higher temperatures are mostly found there.

**Comment 8** — Line 258: "ice droplets" should be "ice crystals"?
**Reply** — Indeed, we adapted the text.

**References**

A. Gettelman, C. G. Bardeen, C. S. McCluskey, E. Järvinen, J. Stith, C. Bretherton, G. McFarquhar, C. Twohy, J. D'Alessandro, and W. Wu. Simulating observations of southern ocean clouds and implications for climate. *Journal of Geophysical Research: Atmospheres*, 125(21): e2020JD032619, 2020. doi: https://doi.org/10.1029/2020JD032619. URL https://agupubs.onlinelibrary.wiley.com/doi/abs/10.1029/2020JD032619.

J. E. Kay, B. R. Hillman, S. A. Klein, Y. Zhang, B. Medeiros, R. Pincus, A. Gettelman, B. Eaton, J. Boyle, R. Marchand, and T. P. Ackerman. Exposing global cloud biases in the community atmosphere model (cam) using satellite observations and their corresponding instrument simulators. *Journal of Climate*, 25(15):5190 – 5207, 2012. doi: 10.1175/JCLI-D-11-00469.1. URL https://journals.ametsoc.org/view/journals/clim/25/15/jcli-d-11-00469.1.xml.

Paul Herenz, Heike Wex, Alexander Mangold, Quentin Laffineur, Irina V Gorodetskaya, Zoë L Fleming, Marios Panagi, and Frank Stratmann. Ccn measurements at the princess elisabeth antarctica research station during three austral summers. *Atmospheric Chemistry and Physics*, 19(1):275–294, 2019.

Kevin Cheuk Hang Sze, Heike Wex, Markus Hartmann, Henrik Skov, Andreas Massling, Diego Villanueva, and Frank Stratmann. Ice nucleating particles in northern greenland: annual cycles, biological contribution and parameterizations. *Atmospheric Chemistry and Physics Discussions*, 2022:1–45, 2022. doi: 10.5194/acp-23-4741-2023. URL https://acp.copernicus.org/articles/23/4741/2023/.

L. Lacher, M. P. Adams, K. Barry, B. Bertozzi, H. Bingemer, C. Boffo, Y. Bras, N. Büttner, D. Castarede, D. J. Cziczo, P. J. DeMott, R. Fösig, M. Goodell, K. Höhler, T. C. J. Hill, C. Jentzsch, L. A. Ladino, E. J. T. Levin, S. Mertes, O. Möhler, K. A. Moore, B. J. Murray, J. Nadolny, T. Pfeuffer, D. Picard, C. Ramírez-Romero, M. Ribeiro, S. Richter, J. Schrod, K. Sellegri, F. Stratmann, B. E. Swanson, E. S. Thomson, H. Wex, M. J. Wolf, and E. Freney. The puy de dôme ice nucleation intercomparison campaign (picnic): comparison between online and offline methods in ambient air. *Atmospheric Chemistry and Physics*, 24(4):2651–2678, 2024. doi: 10.5194/acp-24-2651-2024. URL https://acp.copernicus.org/articles/24/2651/2024/.

P. J. DeMott, A. J. Prenni, X. Liu, S. M. Kreidenweis, M. D. Petters, C. H. Twohy, M. S. Richardson, T. Eidhammer, and D. C. Rogers. Predicting global atmospheric ice nuclei distributions and their impacts on climate. *Proceedings of the National Academy of Sciences*, 107(25):11217–11222, 2010. doi: 10.1073/pnas.0910818107. URL https://www.pnas.org/doi/abs/10.1073/pnas.0910818107.

E. Vignon, S. P. Alexander, P. J. DeMott, G. Sotiropoulou, F. Gerber, T. C. J. Hill, R. Marchand, A. Nenes, and A. Berne. Challenging and improving the simulation of mid-level mixed-phase clouds over the high-latitude southern ocean. *Journal of Geophysical Research: Atmospheres*, 126(7):e2020JD033490, 2021. doi: 10.1029/2020JD033490. URL https://agupubs.onlinelibrary.wiley.com/doi/abs/10.1029/2020JD033490. e2020JD033490 2020JD033490.

---

## Author Response (AR2)

**Response to the reviewers**

Florian Sauerland et al.

August 2024

**Comments to the Editor**

We address all minor change suggestions. Please find a point to point response to the comments of the reviewers underneath. Additionally, we did some minor changes without changing any content, mostly to have a consistent style for numbers and units, which can be found in the track-changes document. We like to thank the reviewers for their continued improvement suggestions and their help in improving the manuscript. In the following, we address their remaining concerns and comments about our work.

**Reviewer 1**

"The comments were well addressed by the authors. There are only a few technical corrections remaining (comment numbers from the author's answer):"

**Comment 2-3** — "It should be "effect on", not 'effect in'."

**Reply** — This has been corrected in the manuscript.

**Comment 12** — "The author's answer states that they have changed the wording, but the revised manuscript and the track-changes file don't show this. Probably the changes weren't saved."

**Comment 13** — "Same as comment 12. The paragraph that should be removed (currently it is appearing double - here and in the beginning of the Results section) is still there."

**Reply** — The missing changes are included in the manuscript now, thank you for noticing. We can only assume that this is the result of a temporary connection loss to the online text editor.

**Comment 22** — In line 346 in the track-changes document, the word "the" appears twice ("comparing the the timesteps"). But actually, I suggest further changes than only removing one "the". I would suggest to change this sentence to the following to make it clearer: "As Fig. 6g shows, the increase in CRE in the VL setting is especially pronounced during timesteps with a smaller CRE." (The words "more" and "comparing" suggest a comparison, but this is not really given in this sentence.)

**Reply** — We adapted the suggested reformulation.

**New comment regarding Fig. 6** — It looks like you put in the wrong plot in Fig. 6g when correcting the figure towards non-italic axis labels (same plot as 5g, PEA instead of SO). Please correct that.

**Reply** — This is indeed an error, so we corrected Figure 6, as well as Figures 2 and 3, where the colorbar for the Cloud Ice Content plots incorrectly said "Cloud Liquid Water Content".

**Reviewer 2**

"The authors have done a conscientious job in responding to the reviews and the manuscript has been improved."

**Comment 1** — "It seems the observational INP data are not being used as a very strong constraint in this work. The authors note that their focus is not on the spectra themselves (the temperature dependence of the INP) but on differences between overall INP concentrations. This comment, about the temperature dependence of the INP spectra not being relevant, seems a little strange, since the contrasting behaviors between relationships in summer and winter were explained by temperature differences."

**Reply** — We adapted the discussion to address this point:

- Nevertheless, our findings with respect to the temperature sensitivity of the cloud response to INP concentration changes should still hold, as the steeper increase in the activation temperature profile measured in campaigns such as MARCUS [Vignon et al., 2021] would only cause an even stronger temperature sensitivity than the gradual increase we used here. In addition to that, the temperatures in winter at the station are low enough to explain the reduced sensitivity by homogeneous nucleation. Only the low sensitivity we found over the Southern Ocean in summer would potentially be affected, as the temperature range found for this situation would mostly fall in the region with a higher increase, but even keeping that in mind, the INP settings we tested covered a wide enough range for any increase from realistic values to fall into the tested range.

**Comment 2** — "I may be missing something, but in comparing the timeline of INP concentrations (now shown as Figure A3) with the simulation time period shown in Figure 2, the cloud seems to form in Figure 2 starting 2012-02-09 through part of 2012-02-11. Does this not coincide with the very last portion of the Figure A3 timeline, where the INP concentrations drop dramatically? Is this drop due to scavenging by the cloud and is that a significant effect on the cloud evolution? The explanation provided for fluctuations earlier in the run is that concentration changes are due to synoptic-scale weather systems, presumably because scavenging of INPs elsewhere in the domain had occurred prior to arrival at the site."

**Reply** — We compared cloud water and ice contents with wind direction and INP concentration and can verify that the INP concentration drop shown in Figure A3 is caused by upstream scavenging, as can be seen by the very low

[Figure]

Figure 1: INP concentration (left) at -20°C and wind direction and strength (right) on 10 February 2012, 00:00h, at a model level corresponding to 2250 m height a.g.l. at PEA (red diamond), in the VL simulation. Please note that only INP concentrations of the fifth temperature bin are shown, i.e. not the sum of temperature bins 1 through 5.

INP concentration in areas corresponding to the areas with strong northerly winds in the Figure underneath. We also added a sentence to the corresponding paragraph to clarify this:

- The drop in concentration towards the end of the shown period can be explained by upstream scavenging of available INPs (not shown).

**Comment 3** — "This is a study aimed at cloud processes, namely, the competition between liquid and ice particle formation and evolution. I encourage the authors to implement further diagnostics into the modeling framework so cloud processes can be more thoroughly probed. In this way, they can ascertain whether secondary ice processes are playing a role in the simulations."

**Reply** — In our paper, we looked into the spatial and temporal evolution of cloud ice, cloud water, and INPs. We acknowledge that further research regarding secondary ice processes that play into ice production is valuable. We herefore suggest this in the conclusions now. However, implementing further diagnostic variables is out of scope for this paper and would require changes to the model as well as new simulations. We keep this feedback in mind for follow-up work.

**References**

E. Vignon, S. P. Alexander, P. J. DeMott, G. Sotiropoulou, F. Gerber, T. C. J. Hill, R. Marchand, A. Nenes, and A. Berne. Challenging and improving the simulation of mid-level mixed-phase clouds over the

high-latitude southern ocean. *Journal of Geophysical Research: Atmospheres*, 126(7):e2020JD033490, 2021. doi: 10.1029/2020JD033490. URL `https://agupubs.onlinelibrary.wiley.com/doi/abs/10.1029/2020JD033490`. e2020JD033490 2020JD033490.

---

## Author Response (AR3)

[revised manuscript text omitted]
 10^{-5}\,\text{L}^{-1}$; L = low INP setting, $5 \times 10^{-3}\,\text{L}^{-1}$; M = medium INP setting, $5 \times 10^{-2}\,\text{L}^{-1}$; H = high INP setting, $2 \times 10^{-1}\,\text{L}^{-1}$, VH = very high INP setting, $2\,\text{L}^{-1}$; all INP concentration settings at $-20\,°\text{C}$.

The radiative effects caused by these changes in cloud phase are small. Figure 5a shows that in general, the median and mean cloud radiative effects generally stay between $50\,\text{W}\text{m}^{-2}$ and $60\,\text{W}\text{m}^{-2}$ for the summer period, with the extremes being slightly lower in the VH setting, and no clear trends connected to INP concentrations. Only the M setting compared to the VH setting shows a statistically significant difference in the paired t-test (Table 3). When the total CRE is split up into a shortwave and longwave part (Fig. 5b and c), the means of the shortwave CRE are decreasing towards a lower INP concentration, indicating that the higher liquid share is more optically thick and therefore reflects a higher portion of sunlight back to space. This is offset by the trend of the longwave CRE, which is increasing towards a lower INP concentration, indicating that the higher liquid portion also reflects more radiation back to the ground. If we, however, only look at the time steps with significant liquid water present (Fig. 5d), we can see that in the VL and L settings, the mean CRE is significantly lower than in the cases with higher INP concentrations (VL: $65.4\,\text{W}\text{m}^{-2}$, L: $67.2\,\text{W}\text{m}^{-2}$; M: $68.8\,\text{W}\text{m}^{-2}$, H: $69.4\,\text{W}\text{m}^{-2}$, VH: $69.7\,\text{W}\text{m}^{-2}$), indicating that for these thicker clouds, the increased shortwave reflection outweighs the longwave reflection. The sample size, however,

[Figure]

**Figure 5.** Different CRE (cloud radiative effects) statistics for the summer (a-f) and winter (g) period, averaged over a 21x21 area around the grid cell of PEA. TQI = vertically integrated cloud ice, TQC = vertically integrated cloud water. The blue markers indicate individual timesteps, the red solid line the median, and the black dashed line indicated the mean. Sample sizes for the subfigures: (a, b, c): 131, (d, e, f): 11, (g): 61. VL = very low INP concentration setting, $1 \times 10^{-5}\,\text{L}^{-1}$; L = low INP setting, $5 \times 10^{-3}\,\text{L}^{-1}$; M = medium INP setting, $5 \times 10^{-2}\,\text{L}^{-1}$; H = high INP setting, $2 \times 10^{-1}\,\text{L}^{-1}$, VH = very high INP setting, $2\,\text{L}^{-1}$; all INP concentration settings 
[revised manuscript text omitted]